EMBO
reports

# Senescent endothelial cells promote pathogenic neutrophil trafficking in inflamed tissues

Loïc Rolas [1,10], Monja Stein[1,10], Anna Barkaway[1], Natalia Reglero-Real[1], Elisabetta Sciacca[2,3], Mohammed Yaseen[1], Haitao Wang[1], Laura Vazquez-Martinez[1], Matthew Golding[1], Isobel A Blacksell[4], Meredith J Giblin [1], Edyta Jaworska [3], Cleo L Bishop[5], Mathieu-Benoit Voisin [1], Carles Gaston-Massuet[6], Liliane Fossati-Jimack[3], Costantino Pitzalis[3], Dianne Cooper[4], Thomas D Nightingale[1], Carlos Lopez-Otin [7,8], Myles J Lewis[2,3] & Sussan Nourshargh [1,9] ✉

## Abstract

**Cellular senescence is a hallmark of advanced age and a major instigator of numerous inflammatory pathologies. While endothelial cell (EC) senescence is aligned with defective vascular functionality, its impact on fundamental inflammatory responses in vivo at single-cell level remain unclear. To directly investigate the role of EC senescence on dynamics of neutrophil-venular wall interactions, we applied high resolution confocal intravital microscopy to inflamed tissues of an EC-specific progeroid mouse model, characterized by profound indicators of EC senescence. Progerin-expressing ECs supported prolonged neutrophil adhesion and crawling in a cell autonomous manner that additionally mediated neutrophil-dependent microvascular leakage. Transcriptomic and immunofluorescence analysis of inflamed tissues identified elevated levels of EC CXCL1 on progerin-expressing ECs and functional blockade of CXCL1 suppressed the dysregulated neutrophil responses elicited by senescent ECs. Similarly, cultured progerin-expressing human ECs exhibited a senescent phenotype, were pro-inflammatory and prompted increased neutrophil attachment and activation. Collectively, our findings support the concept that senescent ECs drive excessive inflammation and provide new insights into the mode, dynamics, and mechanisms of this response at single-cell level.**

**Keywords** Endothelium; Inflammation; Neutrophils; Senescence
**Subject Categories** Molecular Biology of Disease; Vascular Biology & Angiogenesis

## Introduction

Cellular senescence, a hallmark of ageing (López-Otín et al, 2013, 2023), is a state of stable cell cycle arrest induced by a variety of triggers such as DNA damage, oncogene activation, and organelle dysfunction (Schumacher et al, 2021). Providing a protective role, cellular senescence is linked to wound healing, embryogenesis, and tumor suppression (Burton and Krizhanovsky, 2014). However, excessive accumulation of senescent cells contributes to numerous inflammatory conditions, such as metabolic, neurodegenerative, and cardiovascular conditions, and the pathogenesis of various age-related syndromes. Indeed, transplanting senescent cells into young mice causes physical dysfunction (Xu et al, 2018) and therapies targeting senescent cells (e.g., senolytics) are efficacious in preclinical models (Camell et al, 2021; Suda et al, 2021; Xu et al, 2018) and early-stage human clinical trials (Hickson et al, 2019; Justice et al, 2019). Considered a heterogenous phenomenon that is cell- and context-dependent, there is no unifying marker for cellular senescence and as such it is commonly characterized by quantification of multiple biochemical and morphological parameters. These include expression of cyclin-dependent kinase inhibitors, measurement of DNA damage, and activity of senescence-associated β-galactosidase (SA-β-gal). Critically, a prominent feature of senescent cells is their increased capacity to secrete a wide range of pro-inflammatory mediators, such as cytokines, chemokines, growth factors, and proteases. This response is termed the senescence-associated secretory phenotype (SASP) and likely plays a prominent role in mediating key pathological effects of senescent cells (Coppé et al, 2010).

While cellular senescence is linked to dysregulated innate and adaptive immunity in a cell-autonomous (e.g., via immune cell senescence) or non-autonomous (e.g., via SASP) manner (Lee et al, 2022), the specific impact of this phenomenon on neutrophil trafficking into inflamed tissues requires further explorations. In

[1]Centre for Microvascular Research, William Harvey Research Institute, Faculty of Medicine and Dentistry, Queen Mary University of London, London, UK. [2]Centre for Translational Bioinformatics, William Harvey Research Institute, Faculty of Medicine and Dentistry, Queen Mary University of London, London, UK. [3]Centre for Experimental Medicine and Rheumatology, William Harvey Research Institute, Faculty of Medicine and Dentistry, Queen Mary University of London, London, UK. [4]Centre for Biochemical Pharmacology, William Harvey Research Institute, Faculty of Medicine and Dentistry, Queen Mary University of London, London, UK. [5]Centre for Cell Biology and Cutaneous Research, Blizard Institute, Faculty of Medicine and Dentistry, Queen Mary University of London, London, UK. [6]Centre for Endocrinology, William Harvey Research Institute, Faculty of Medicine and Dentistry, Queen Mary University of London, London, UK. [7]Centre de Recherche des Cordeliers, Inserm U1138, Université Paris Cité, Sorbonne Université, Paris, France. [8]Facultad de Ciencias de la Vida y la Naturaleza, Universidad Nebrija, Madrid, Spain. [9]Centre for Inflammation and Therapeutic Innovation, Queen Mary University of London, London, UK. [10]These authors contributed equally: Loïc Rolas, Monja Stein. ✉E-mail: s.nourshargh@qmul.ac.uk

providing crucial and rapid host defence, neutrophils are among the first immune cells to infiltrate sites of injury and infection (Ley et al, 2007; Nourshargh and Alon, 2014). Abnormal recruitment and/or activation of neutrophils is, however, a key feature of numerous acute and chronic inflammatory conditions (Mayadas et al, 2014; Silvestre-Roig et al, 2020) and is associated with higher incidence of multiple ageing-associated disorders, such as stroke, acute lung injury, sepsis, and cardiovascular conditions (Gullotta et al, 2023; Kwok et al, 2023; Simmons et al, 2021; Van Avondt et al, 2023). To migrate into inflamed tissues, neutrophils are required to interact with venular endothelial cells (ECs) that line the inner aspect of all blood vessels and provide the principal barrier to cellular and non-cellular blood constituents. In response to an inflammatory insult, ECs are stimulated to express critical pro-adhesive and pro-migratory molecules that collectively facilitate a cascade of luminal neutrophil-EC interactions. This includes neutrophil rolling, firm arrest and crawling on ECs, responses that are prerequisites to full breaching of the venular wall (Nourshargh and Alon, 2014). We have previously shown that disrupted expression or localization of key EC adhesion/activation molecules can perturb the dynamics of neutrophil-venular wall interactions from a physiological mode towards a pathogenic state (Barkaway et al, 2021; Colom et al, 2015; Girbl et al, 2018; Owen-Woods et al, 2020; Reglero-Real et al, 2021; Woodfin et al, 2011). This was most notable following aberrant presentations of venular chemokines (e.g., CXCL1) (Girbl et al, 2018; Owen-Woods et al, 2020), a profile dramatically observed in inflamed aged tissues (Barkaway et al, 2021). In the light of mounting evidence that EC senescence can contribute to compromised vascular responses in aged and inflamed organs (Jia et al, 2019; Ting et al, 2021), here we sought to directly investigate the impact of EC senescence on neutrophil trafficking. For this purpose, we employed a mouse model of EC senescence based on the genetics of the fatal premature aging disorder Hutchinson-Gilford Progeria Syndrome (HGPS). HGPS, a progeroid laminopathy, is caused by a point mutation in the *LMNA* gene that results in the production of progerin, a truncated form of the nuclear protein lamin A. At the molecular level, progerin causes numerous senescence-associated responses such as genomic instability, DNA damage and nuclear deformation and as such affects multiple ageing-related processes such as mTOR signaling, inflammation, microRNA activation, and stress response mechanisms (Cenni et al, 2020). In line with this, the dramatic cellular phenotypes in HGPS include indicators of premature senescence (Cenni et al, 2020), and as found in physiologically aged mice, agents that target senescent cells extend the lifespan of progeroid animals (Camell et al, 2021; Suda et al, 2021; Xu et al, 2018). Crucially, progerin is expressed at low levels in normal subjects (e.g., in atherosclerotic plaques) (Mcclintock et al, 2007; Olive et al, 2010) and there is evidence of a synergistic relationship between telomere dysfunction and progerin production during the initiation of cellular senescence (Cao et al, 2011). Here, we report that mice with conditional expression of progerin in ECs exhibit senescent endothelia with pro-inflammatory characteristics that support excessive neutrophil attachment to the luminal aspect of venules and mediate neutrophil-dependent vascular permeability in models of inflammation. Mechanistically, through the application of high resolution intravital microscopy, we provide unequivocal evidence for senescent progerin-expressing ECs being overtly pro-adhesive for neutrophils in a cell autonomous manner that was CXCL1-dependent. The results link EC senescence with development of dysregulated neutrophil-venular wall interactions and inflammation.

## Results

### Establishment and characterization of a progerin-driven EC senescence mouse model

As a key feature of HGPS is premature cellular senescence, a response molecularly aligned with an accelerated version of physiological senescence (Cenni et al, 2020), we established mice that selectively express progerin in ECs as a mouse model of vascular senescence. Briefly, knock-in mice carrying an internal LoxP-stop-LoxP-1827C > T p.Gly609Gly allele in the *Lmna* gene (*Lmna*^*LCS/LCS*^) (Osorio et al, 2011) were crossed with the *Tie2-Cre* transgenic strain to generate mice expressing progerin in ECs (*Tie2-Cre;Lmna*^*LCS/LCS*^*;Rosa26*^*tdTomato/tdTomato*^) (Fig. 1A). Animals expressing Cre recombinase, and hence progerin, are hereon referred to as Cre⁺, and their littermate controls, as Cre⁻. Combining the *Rosa26-CAG-LoxP-stop-LoxP-tdTomato* (tdTmt) Cre reporter onto the *Lmna* background provided a visual means of identifying progerin-expressing ECs (referred to as tdTmt⁺) and those that do not (tdTmt⁻) by confocal microscopy (Fig. 1B). With this strategy we noted that in this model, ~60% of ECs expressed tdTomato fluorescence in Cre⁺ mice (Fig. EV1A), a recombination profile that gave rise to a mosaic tdTomato expression pattern in the microcirculation of all tissues analyzed (Fig. 1B). As anticipated, no tdTomato was detected in blood vessels of Cre⁻ animals (Fig. EV1A,B). To categorically ascertain the correlation between progerin and tdTomato expression in ECs, we sought to immunostain venular segments for progerin. For this purpose, and due to the lack of a commercially available specific anti-mouse progerin antibody (Ab), we developed a novel rabbit polyclonal Ab (pAb). Here, as antigen, we used the unique spliced peptide sequence of mouse progerin, GAQSSQNC, which exhibits a single amino-acid difference from its human homolog (Fig. 1C). Immunoblotting experiments confirmed that our newly developed pAb detected a strong signal at the expected molecular weight of progerin (~70 kDa) in FACS-sorted Cre⁺ (but not Cre⁻) mouse lung endothelial cells (MLECs) (Fig. EV1C). Furthermore, immunofluorescence (IF) staining and confocal microscopy of Cre⁺ cremasteric venules showed a clear and specific EC nuclear immunoreactivity of the pAb (Fig. 1D). Importantly, IF analysis of cremaster muscle tissues of Cre⁺ mice detected no progerin in tdTmt⁻ ECs while all tdTmt⁺ ECs were stained for progerin (Fig. 1D,E and Movie EV1). This 100% correlation between progerin and tdTomato expression was also noted in ECs of ear skin and lungs (Fig. 1F–H). Consequently, hereafter, tdTmt⁺ and tdTmt⁻ ECs are referred to as Prog⁺ and Prog⁻ ECs, respectively. Collectively, these results illustrated the specificity of our pAb for mouse progerin, a critical feature that has also been validated by our collaborators (Marcos-Ramiro et al, 2021; Santiago-Fernández et al, 2019; Sánchez-López et al, 2021). Of note, relevant to the objectives of the present study, IF analysis of inflamed cremaster muscles showed no evidence of progerin expression in luminal or extravasated neutrophils (Fig. EV1D). Together, these data confirm the robust nature of our model of EC progerin expression that additionally and uniquely facilitates direct comparison of Prog⁺

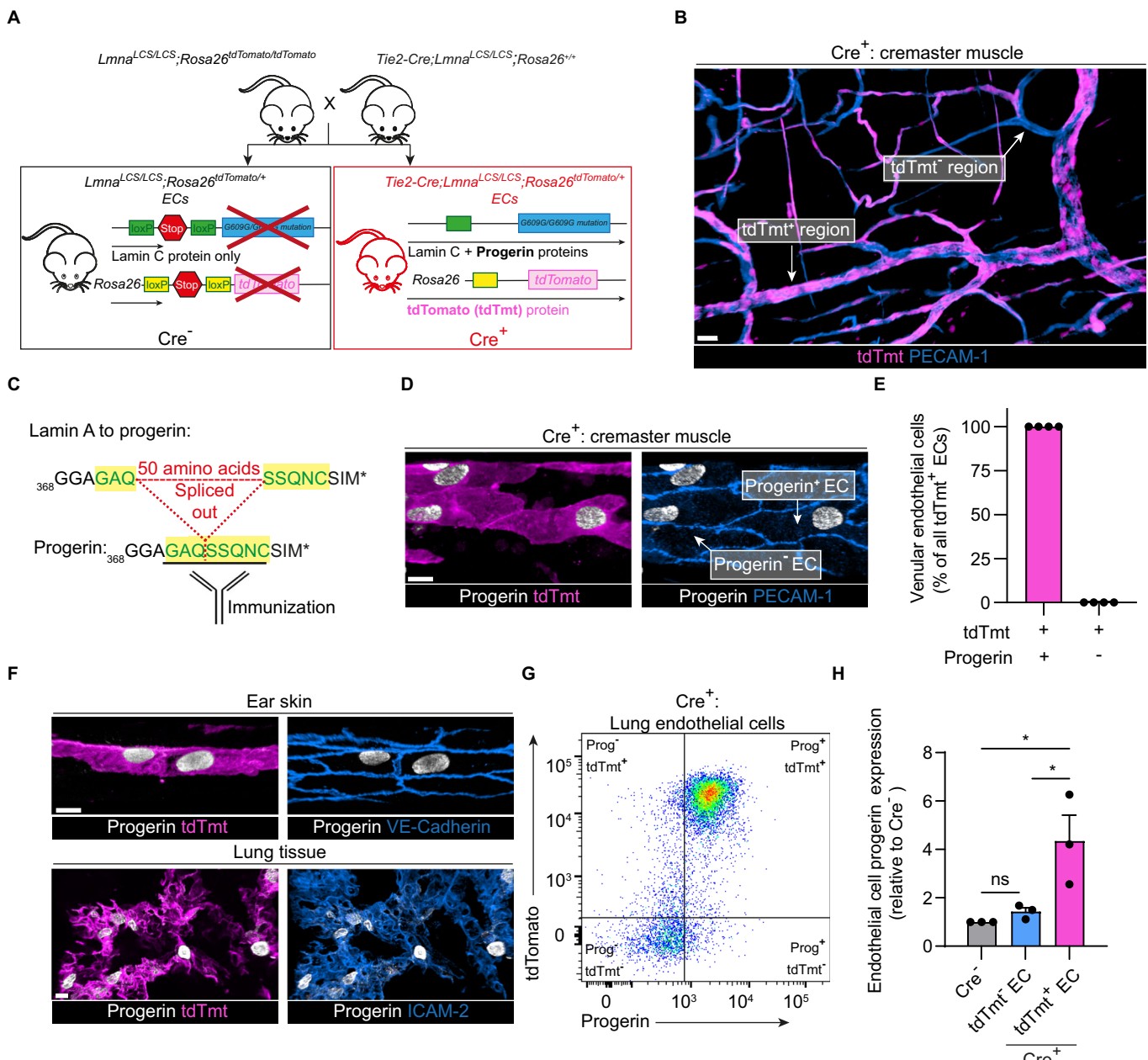

**Figure 1. Development of mouse model of vascular aging with mosaic distribution of progerin expression in endothelial cells.**

(A) Schematic depicting the generation of the *Tie2-Cre;Lmna^LCS/LCS^;Rosa26^tdTomato/+^* mouse whereby progerin and tdTomato fluorescent protein are conditionally expressed in ECs. Progerin and tdTomato-expressing mice are referred to as "Cre^+^" and their littermate negative controls as "Cre^−^". (B) The cremasteric microvasculature of a Cre^+^ mouse was immunostained for PECAM-1 (CD31) and analyzed by confocal microscopy tile scanning, illustrating the mosaic pattern of tdTomato expression; regions with or without tdTomato fluorescence are indicated. Scale bars: 20 μm. (C) Schematic depicting the sequence of the mouse progerin peptide that was used to generate the novel specific rabbit anti-mouse progerin Ab. (D–G) Tissues of Cre^+^ animals were co-immunostained for progerin and other EC markers, as shown. (D, E) Representative confocal images of cremaster muscle post-capillary venules depicting the coupling of progerin and tdTomato (tdTmt) expression (Scale bar: 10 μm) and the quantification of this association (*n* = 4 mice/group in 1 experiment). (F) Representative confocal images depicting coupling between endothelial cell progerin and tdTomato expression in ear and lung tissues. Scale bars: 10 μm. (G, H) Representative flow cytometry plots depicting the correlation of progerin and tdTomato expression in lung endothelial cells of Cre^+^ mouse and its quantification (*n* = 3 mice/group in 2 independent experiments). Data information: (E) Data are presented as mean. (H) Data are mean ± SEM. *P < 0.05 and ns: not significant. One-way ANOVA followed by Tukey's post hoc test. Source data are available online for this figure.

and Prog^−^ ECs from the same animal, and indeed within the same venular segment.

Next, we sought to characterize the senescence and pro-inflammatory state of progerin-expressing ECs as compared to

progerin-deficient cells. Initially we immunostained ear skin of Cre^+^ mice for phosphorylated histone H2AX (γH2AX) as a marker of DNA double-strand breaks. Taking advantage of the mosaic progerin expression, the γH2AX immunostaining revealed a low

punctate expression pattern in the nuclei of Prog⁻ ECs that was significantly elevated in adjacent Prog⁺ ECs (Fig. 2A,B). Indicating enhanced DNA damage in progerin-expressing ECs, we next analyzed Prog⁺ MLECs for multiple other markers of cellular senescence. Specifically, Prog⁺ as compared to Prog⁻ MLECs exhibited significantly elevated levels of SA-β-Galactosidase activity (Fig. 2C,D) and morphological changes, such as increased cell size (Figs. 2E and EV1E) and granularity (Figs. 2F and EV1F) when analyzed by flow cytometry. Imaging of whole-mount cremaster muscles additionally revealed abnormal nuclear morphology in ECs of progerin-expressing tdTmt⁺ cells (Fig. 2G,H). Finally, while we did not detect notable changes in cell surface protein expression of key EC adhesion molecules at steady state (Fig. EV1G), VCAM1 and ICAM1 mRNA levels were upregulated in Prog⁺ MLECs (Fig. 2I,J), suggesting a pro-inflammatory state.

Overall, we report on the establishment of valuable and robust tools for investigations of progerin expression and progerin-driven senescence in microvascular ECs in vivo.

## Senescent endothelial cells are pro-adhesive for neutrophils in vivo

While aligned with vascular abnormalities, the role of senescent ECs in immune cell migration and inflammation requires further exploration. To address, we took advantage of the mosaic pattern of Cre recombinase expression in the cremaster muscle microcirculation of our Cre⁺ mice (Figs. 1D and 3A) and directly investigated the impact of senescent ECs on neutrophil trafficking. Initially, analysis of whole-mounted IL-1β-stimulated cremaster muscles indicated a greater localization of neutrophils to vascular regions expressing tdTmt⁺ ECs (both intra- and extra-luminally), as compared to tdTmt⁻ microvascular segments (Fig. 3A,B and Movie EV2). However, quantification of neutrophil extravasation revealed no statistical difference between IL-1β-stimulated Cre⁻ and Cre⁺ mice (Fig. 3B). To get insight to the dynamics of neutrophil interactions with inflamed senescent progerin-expressing venules in real-time, we employed high resolution confocal intravital microscopy (IVM). This was applied to IL-1β-stimulated cremaster muscles of *Tie2-Cre;Lmna^{LCS/LCS};Rosa26^{tdTomato/tdTomato}* mice intercrossed with *Lyz2-EFGP-ki* mice for tracking of GFP^{high} neutrophils. Within this model, cremaster venular ECs were labeled via locally applied fluorescently-tagged non-blocking anti-PECAM-1 mAb, as previously described (Barkaway et al, 2021; Reglero-Real et al, 2021). Taking advantage of the patchy expression profile of tdTmt in Cre⁺ mice, we noted a significantly greater attachment of neutrophils to Prog⁺ ECs (Fig. 3C,D and Movie EV2). Similarly, the time elapsing between initial neutrophil arrest to detachment, or initiation of crawling, was significantly longer in Cre⁺ venular segments comprising Prog⁺ ECs (Fig. 3E). Further, more crawling neutrophils and with slower speed, were observed on Prog⁺ ECs than on Prog⁻ ECs within the same venules of Cre⁺ mice (Fig. 3F,G). Intriguingly, neutrophils that crawled sequentially on a tdTmt⁺ and an adjacent tdTmt⁻ EC (8 neutrophils tracked in total over 4 mice), exhibited an abrupt increased crawling velocity as they moved from the surface of Prog⁺ (~3.8 μm/min ±0.6 SEM) to Prog⁻ ECs (~5.4 μm/min ±0.5 SEM) (Fig. 3H and Movie EV3). This observation categorically demonstrates that increased neutrophil attachment to inflamed venular walls of Cre⁺ mice is strictly localized to progerin-expressing cells. In line with the noted erratic motility of neutrophils

on progerin-expressing ECs, neutrophils crawling in Cre⁺ venules exhibited greater meandering and reduced directional motility as compared to responses detected in Cre⁻ animals (Fig. EV2A–C). Of note, Cre⁺ animals that did not express the tdTmt transgene, similarly exhibited enhanced neutrophil attachment to stimulated Prog⁺ ECs as compared to Prog⁻ ECs (Fig. EV2D,E).

Collectively these results provide direct evidence for the ability of senescent progerin-expressing ECs to influence the dynamics and profile of neutrophil-venular wall interactions in vivo.

## Senescent endothelial cells support neutrophil-dependent microvascular leakage

Having observed excessive and prolonged attachment of neutrophils to progerin-expressing ECs, we next analyzed vascular leakage as a potential functional implication of this response. While IL-1β did not promote significant vascular leak in cremaster muscles of Cre⁻ mice (quantified by extravascular accumulation of i.v. 20 nm fluorescent beads), this was significantly increased in Cre⁺ animals (Figs. 4A and EV3A). Similar results were obtained when microvascular leakage was quantified by measuring interstitial accumulation of i.v. injected plasma protein tracer TRITC-dextran (MW ~ 75 kDa) (Fig. EV3B). Furthermore, as found with enhanced neutrophil attachment (Fig. EV2D,E), Cre⁺ mice that did not carry the tdTmt transgene exhibited increased IL-1β-induced vascular permeability, as compared to Cre⁻ litter mates (Fig. EV3C). Collectively these results illustrate that tdtomato expression does not contribute to the pro-inflammatory state of inflamed EC progerin-expressing tissues. Of note, highly recombined venular segments (>70% of ECs being Prog⁺) were associated with a 3.8-fold increase in microbead extravasation as compared to venular segments with lower recombination frequency (<30% of ECs being Prog⁺) (Fig. 4B). Crucially, mice depleted of their circulating neutrophils showed complete inhibition of microbead extravasation across inflamed Cre⁺ cremasteric venules (Fig. 4C). Furthermore, in investigating another vascular bed and additional inflammatory stimuli, vascular leakage in the mouse dorsal skin as induced by intradermal injection of the neutrophil chemoattractant LTB₄ was similarly enhanced in Cre⁺ mice, as compared to Cre⁻ controls (Fig. 4D). Conversely, in both skin and cremaster muscle, the direct acting vascular permeability inducing agent histamine elicited comparable leakage responses in Cre⁻ and Cre⁺ animals (Figs. 4D and EV3D). Together, these results indicate that excessive attachment of neutrophils to senescent progerin-expressing ECs leads to compromised barrier function of inflamed venules.

## Endothelial cell progerin promotes neutrophil accumulation and vascular leakage in a model of lung damage

To investigate the impact of EC senescence in a pathological inflammatory setting, EC progerin-expressing mice were subjected to a model of acute lung injury. For this purpose, mice were injected intraperitoneally with lipopolysaccharide (LPS) and peptidoglycan (PepG), or PBS as control, and lung neutrophil infiltration and vascular leakage were quantified. Within this model of endotoxemia, LPS/PepG exerted a profound inflammatory response in the lungs of all animals tested. However, 4 h post induction of the model, Cre⁺ mice exhibited a significantly elevated

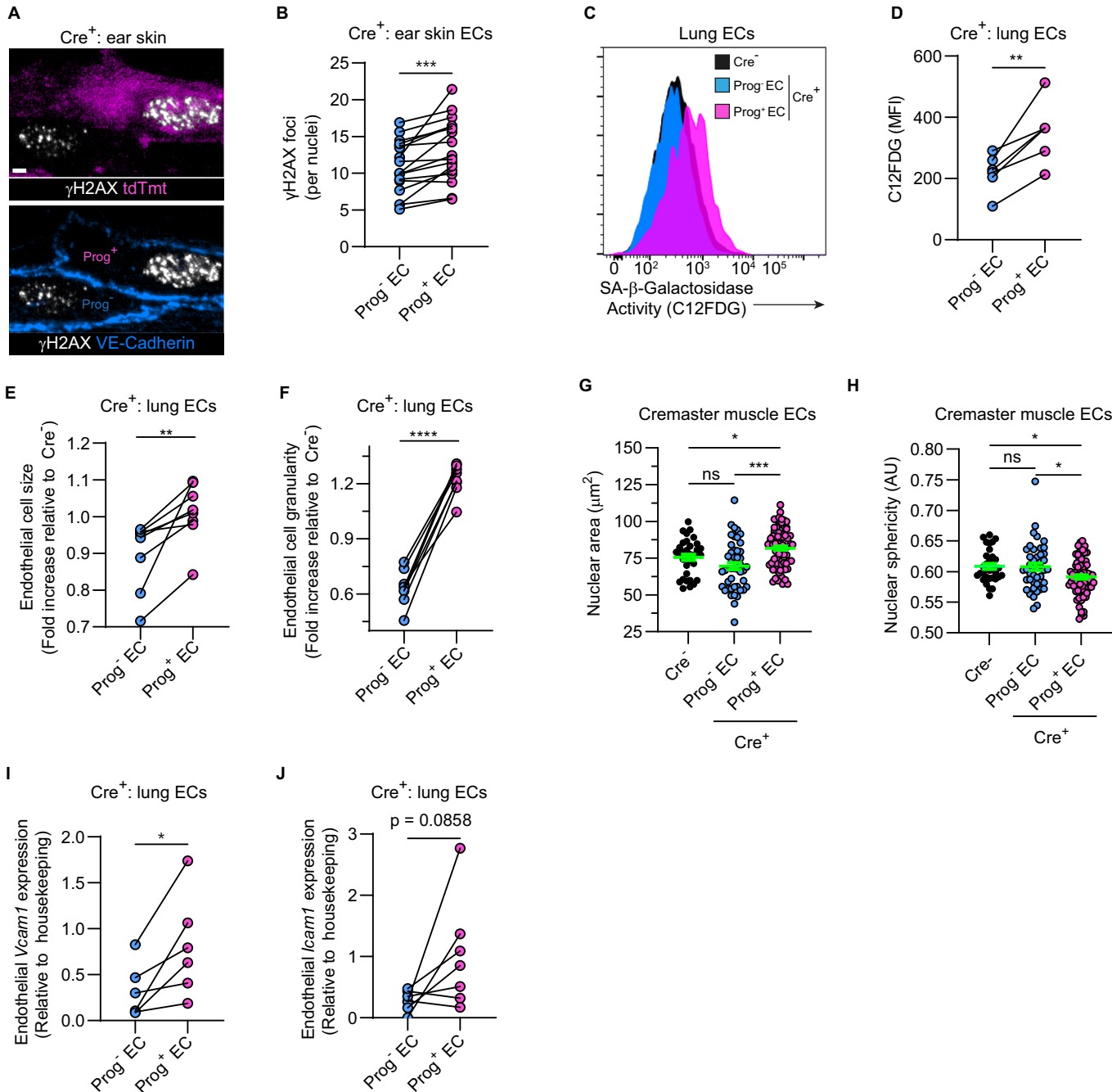

**Figure 2. Progerin-expressing endothelial cells are senescent and exhibit a pro-inflammatory phenotype.**

(A, B) Mouse ear skin was immunostained for VE-cadherin (to delineate ECs) and γH2AX (DNA damage marker). (A) Representative confocal images depicting the correlation between tdTomato and γH2AX expression in ECs of Cre+ mice. Scale bar: 3 μm. (B) Quantification of the number of γH2AX positive foci per EC nucleus, each pair of linked blue and magenta data points represent one Cre+ mouse ($n = 17$ mice in 7 independent experiments). (C–F) Lung ECs from Cre- and Cre+ mice were analyzed for multiple senescence markers by flow cytometry ($n = 6$ mice/group in 2 independent experiments). (C) Representative flow cytometry histogram plots depicting C12FDG fluorescence intensity and (D) its quantification as mean fluorescence intensity (MFI) per mouse. (E, F) Quantification of the EC mean cell size (FSC-A) and granularity (SSC-A), respectively, of progerin positive and negative lung ECs of Cre+ mice as expressed per mouse, relative to ECs of Cre- animals. (G, H) Cremaster microvascular ECs from Cre- and Cre+ mice were imaged by confocal microscopy and their (G) nuclear surface area and (H) nuclear sphericity were quantified. Each data point represents a single-cell nucleus ($n = 3$–4 mice/group in 3 independent experiments). (I, J) Icam1 (I) and Vcam (J) gene expression levels in progerin positive and negative lung ECs of naïve Cre+ mice sorted from the same mice were assayed by RT-qPCR ($n = 6$–7 mice/group in 3 independent experiments). Data information: (B, D–F, I, J) *$P < 0.05$; **$P < 0.01$, ***$P < 0.001$; ****$P < 0.0001$. Analysis by two-tailed paired Student's $t$-test. (G, H) Data are presented as mean ± SEM. *$P < 0.05$; ***$P < 0.001$; ns: not significant. One-way ANOVA followed by Tukey's post hoc test. Source data are available online for this figure.

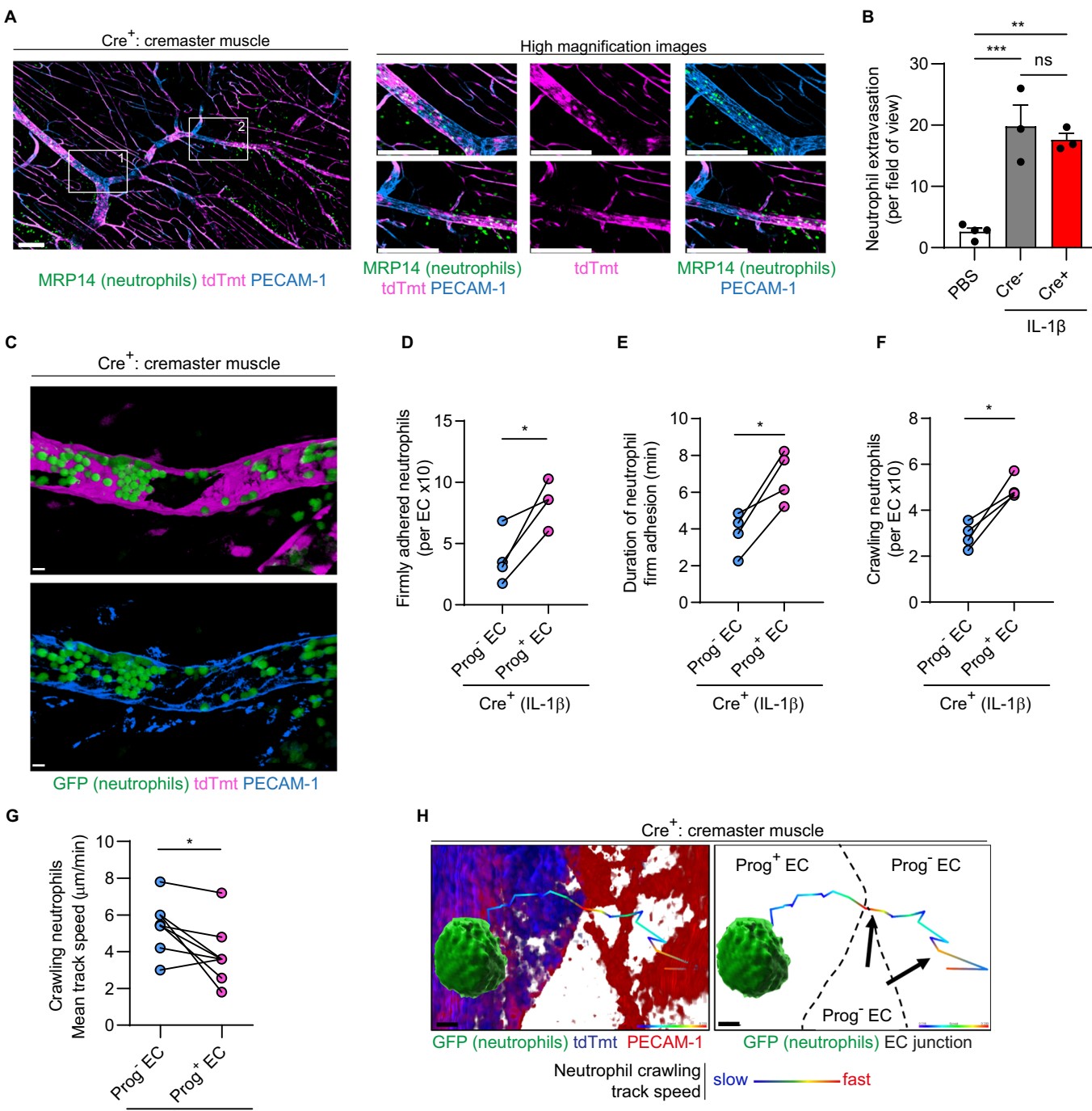

**Neutrophil crawling track speed** | slow ━━━━━ fast

neutrophil accumulation response and compromised vascular integrity, as compared to Cre⁻ animals (Fig. 5A,B). Importantly, we detected significantly higher infiltration of neutrophils in close apposition to Prog⁺ (tdTmt⁺) ECs as compared to Prog⁻ (tdTmt⁻) ECs (Fig. 5C,D). Since similar results were observed in the context of vascular leakage (Fig. EV3E), and in line with the findings from inflamed cremaster muscles, these results suggest a direct role for senescent progerin-expressing ECs in development of dysregulated pathological inflammation. Noting a positive correlation between enhanced vascular permeability and number of neutrophils in Cre⁺

lung tissues (Fig. 5E), the findings support the notion of a causal link between these inflammatory events.

## Inflamed progerin-expressing endothelial cells show elevated levels of CXCL1

To shed light on the molecular changes induced by senescent ECs at a whole organ level, we conducted targeted transcriptomic analyses on lungs acquired from Cre⁻ and Cre⁺ mice treated with i.p. PBS or LPS/PepG. For this purpose, total RNA was extracted from lungs and

Figure 3. Microvascular progerin-expressing ECs support enhanced neutrophil adhesion.

Cremaster microvasculature of Cre⁻ and Cre⁺ or *LyzM-EGFP-ki* Cre⁺ mice were acutely inflamed (IL-1β) and immunostained for PECAM-1 (ECs) with tdTomato fluorescence identifying progerin expressing endothelial cells. (A) Representative confocal image showing neutrophil (MRP14⁺) accumulation in close apposition to tdTomato expressing ECs. White boxes delineate magnified ROIs in the right panel. Scale bars: 100 μm. (B) The number of extravasated neutrophils per field of view was quantified in control (PBS) or IL-1β-stimulated Cre⁻ and Cre⁺ animals (n = 3–4 mice/group in 3 independent experiments). (C–H) Neutrophil behaviors were analyzed and quantified by confocal IVM in *LyzM-EGFP-ki* Cre⁺ mice. (C) Representative confocal image of a stimulated post-capillary venule showing adhesion of EGFP-expressing neutrophils to tdTmt positive ECs. Scale bar: 10 μm. (D) Neutrophil firm adhesion per progerin negative or positive ECs in the same vessel segment, (E) duration of neutrophil firm adhesion to progerin-positive versus negative ECs in the same venular segment, (F) number of crawling neutrophils per progerin negative or positive EC in the same vessel segment. For (B–D), n = 4 mice in 4 independent experiments. (G) Change in neutrophil crawling speed when moving from progerin negative to progerin-positive ECs, each linked pair of data points represents one neutrophil (n = 4 mice in 4 independent experiments). (H) Representative confocal image depicting a neutrophil crawling track from a progerin-positive (tdTmt⁺) to a progerin-negative (tdTmt⁻) endothelial cell. Track color indicates crawling velocity and black arrows show time frames with increased neutrophil velocity. Scale bar: 4 μm. Data information: (B) Data are mean ± SEM. **$P < 0.01$, ***$P < 0.001$, ns: not significant. One-way ANOVA test followed by Tukey's post hoc test. (D–G) *$P < 0.05$. Two-tailed paired Student's t-test. Source data are available online for this figure.

subjected to NanoString gene expression analysis (NanoString nCounter SPRINT system), using a gene expression panel that analyses 785 murine genes involved in multiple inflammatory processes. Initially, analysis of Cre⁻ samples acquired from PBS and LPS/PepG-treated mice revealed upregulation of 257 genes and downregulation of 272 genes (Fig. EV4A). This included upregulation of *Ifit1*, *Rsad2*, *Cxcl2*, *Cxcl11*, and *Sele*, and downregulation of *Ahr*, *Ackr4*, *Cx3cr1*, *Ccr7*, and *Cd79b*. Focusing on the LPS/PepG data, and comparing Cre⁻ and Cre⁺ animals, we detected 90 upregulated genes (e.g., *Cxcl1*, *Ccl2*, *Ccl3*, *Elane*, and *Il6*) and 25 that were downregulated (e.g., *Ccr2*, *Mmp9*, *Cxcr4*, and *Mgam*) (Figs. 6A and EV4B). Pathway enrichment analysis of these data identified classic myeloid responses to inflammation, such as those related to cell migration and antimicrobial responses (Fig. EV4C). Of interest, no significant differences were found between PBS-treated Cre⁻ and Cre⁺ mice (Fig. EV4D), indicating that EC progerin by itself does not drive tissue-level pro-inflammatory molecular changes.

As *Cxcl1* was one of the most upregulated genes in inflamed lungs of Cre⁺ mice (Fig. 6A), and since we previously identified EC-derived CXCL1 as a key regulator of neutrophil firm arrest (Girbl et al, 2018), we hypothesized that elevated protein levels of EC CXCL1 may contribute to the dysregulated inflammatory response of EC progerin-expressing mice. To explore, we quantified CXCL1 protein expression on lung and cremaster ECs of Cre⁻ and Cre⁺ mice by IF under control and inflamed conditions. In lungs, while EC CXCL1 levels were similar between PBS-treated mice of both genotypes, this was significantly elevated in LPS/PepG-treated Cre⁺ animals, as compared to Cre⁻ (Fig. 6B). Similar results were obtained in IL-1β-stimulated cremaster muscles (Fig. 6C). Furthermore, in both tissues we noted greater association of CXCL1 to Prog⁺ ECs as compared to Prog⁻ ECs (Figs. 6D,E and EV4E). Of note, on analyzing protein levels of key EC adhesion molecules, we detected no difference in expression of ICAM-1 or VCAM-1 on stimulated lung Prog⁺ ECs versus Prog⁻ ECs (Fig. EV4F,G). A modest, but significant, increased level of E-selectin was, however, noted on Prog⁺ ECs (Fig. EV4H).

Together, we show that EC progerin confers an exaggerated pro-inflammatory response in inflamed tissues and identify CXCL1 as a significantly expressed molecule on stimulated progerin-expressing ECs.

## Blockade of CXCL1 suppresses inflammatory responses induced by senescent progerin-expressing ECs

Next, we sought to investigate the potential role of EC CXCL1 as a causal driver of dysregulated inflammatory responses in EC progerin-expressing mice. To test this, Cre⁻ and Cre⁺ animals were infused intravenously with a blocking anti-CXCL1 mAb, or an isotype control mAb, and neutrophil-EC interactions in IL-1β-stimulated cremasteric venules were analyzed by confocal IVM. In line with our previous data (Fig. 3), Cre⁺ mice treated with the control mAb exhibited a significantly greater neutrophil attachment to and prolonged firm arrest on Prog⁺ ECs, as compared to Prog⁻ ECs or ECs of Cre⁻ mice (Fig. 7A,B). This elevated response was totally suppressed in mice treated with the anti-CXCL1 mAb, going down to levels detected in Prog⁻ ECs (Fig. 7A,B). Furthermore, in the same model, treatment of mice with the anti-CXCL1 mAb inhibited the enhanced microvascular leakage response of Cre⁺ mice (Fig. 7C). Similar results were obtained in inflamed lungs (Fig. 7D). In line with the low level of CXCL1 detected on ECs of PBS-treated tissues (Fig. 6B,C), no significant neutrophil adhesion was noted in PBS-treated venules of Cre⁻ or Cre⁺ mice (an average of <1 neutrophil/100 μm venular segment, e.g., 0.33 ± 0.17 and 0.38 ± 0.10, in Cre⁻ and Cre⁺, respectively, n = 3–4 mice). Together, these data identify enhanced CXCL1 expression on ECs as a causal instigator of dysregulated inflammatory responses induced by EC progerin. Crucially, analysis of inflamed cremasteric venules of physiologically aged mice revealed a higher level of neutrophil attachment to ECs with a senescent phenotype (p21^high), as compared to p21^low ECs (Fig. 7E,F). Furthermore, p21^high aged ECs exhibited significantly elevated expression of CXCL1 protein (Fig. 7G) and treatment of aged mice with a blocking anti-CXCL1 mAb suppressed the excessive neutrophil adhesion to IL-1β-stimulated venules (Fig. 7H). Collectively, these results illustrate the functional importance of elevated EC-derived CXCL1 in mediating the pro-inflammatory effects of senescent ECs.

## Adhesion to senescent progerin-expressing human ECs induces neutrophil activation

In a final series of experiments, we extended our studies to human cells and sought to determine if senescent progerin-expressing human ECs promoted greater neutrophil adhesion and activation. For this purpose, we established a lentiviral transduced method of expressing GFP-tagged human progerin or lamin A (control) in human umbilical vein endothelial cells (HUVECs). Immunofluorescence and immunoblotting strategies confirmed overexpression of GFP-lamin A and GFP-progerin in transduced HUVECs (Figs. 8A and EV5A). Notably, with both methods, our custom made pAb selectively detected progerin, confirming its cross-

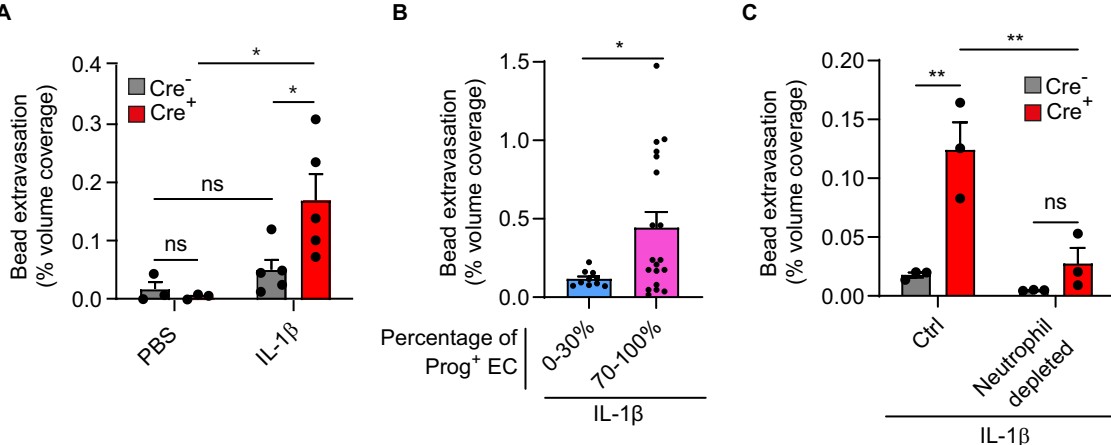

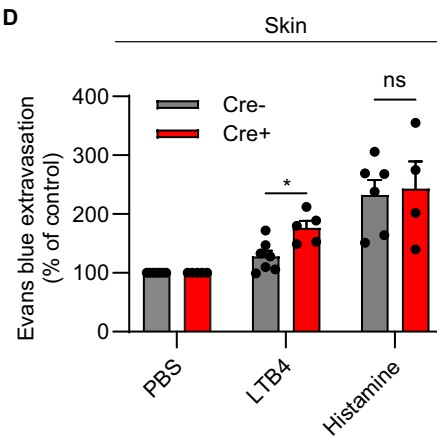

**Figure 4. EC progerin promotes neutrophil-dependent vascular leakage.**

Cremaster microvasculature of Cre⁻ and Cre⁺ mice were acutely inflamed (IL-1β) and immunostained for PECAM-1 (ECs) and MRP14 (neutrophils) with tdTomato fluorescence identifying progerin expressing endothelial cells. (A–C) Quantification of microvascular leakage calculated as percentage coverage of extravasated fluorescent beads (percentage coverage); (A) in Cre⁻ vs Cre⁺ control and IL-1β-stimulated tissues ($n = 3$-5 mice/group in 7 independent experiments); (B) in relation to the percentage of progerin-positive ECs in vessel segments of Cre⁺ mice ($n = 4$ mice/group in 4 independent experiments); and (C) in IL-1β-stimulated control and neutrophil depleted (post treatment with an isotype control or anti-GR-1 antibody, respectively) Cre⁻ and Cre⁺ mice ($n = 3$ mice/group in 3 independent experiments). (D) Vascular leakage in the dorsal skin of Cre⁻ or Cre⁺ mice injected i.d. with PBS, LTB₄ (1 h) or histamine (30 min) ($n = 4$–7 mice/group in 7 independent experiments). Data information: Data are mean ± SEM. *$P < 0.05$; **$P < 0.01$; ns: not significant. (A, C, D) Two-way ANOVA followed by Tukey's (A, C) and Šídák's (D) post hoc test. (B) Two-tailed unpaired Student's *t*-test. Source data are available online for this figure.

reactivity with human as well as mouse progerin. Engineering the HUVEC model to exhibit a mosaic pattern of Prog⁺ ECs across monolayers (~50% of cells being GFP⁺) enabled us to directly compare the biological and morphological features of progerin-expressing, versus non-transduced cells in the same cultures. Guided by our in vivo data and published works (Bidault et al, 2020; Xu et al, 2022), we sought to quantify numerous features of cellular senescence. Here, while control untransduced or GFP-lamin A-transduced cells exhibited a low number of SA-β-galactosidase positive cells, this was significantly increased in GFP-progerin-expressing HUVECs (Fig. EV5B,C). Progerin-expressing HUVECs were additionally larger than control cells and presented multiple nuclear abnormalities such as reduced sphericity and increased size (Figs. 8A,B and EV5D,E). To probe for the existence of a secretory phenotype, we analyzed the conditioned medium of unstimulated and IL-1β-stimulated transduced cells for

~40 different cytokines and chemokines using a multiplex antibody array. Under both basal conditions and following IL-1β stimulation, we detected elevated levels of numerous neutrophil stimulating factors, including the chemokines CXCL8 (IL-8) and CXCL1 in media of GFP-progerin-expressing HUVECs, as compared to those from GFP-lamin A-expressing cells (Fig. 8C). As CXCL8 is the human functional homolog of murine CXCL1, an enhanced level of CXCL8 secretion from progerin-expressing ECs was additionally confirmed by ELISA (Fig. 8D). These results provide direct evidence for progerin-expressing ECs acting as a source of neutrophil stimulatory/chemoattractant molecules during inflammation.

Having established and characterized the senescent state of our human progerin-expressing EC model, we investigated the ability of these cells to support neutrophil adhesion and activation using freshly isolated human blood neutrophils. While neutrophil

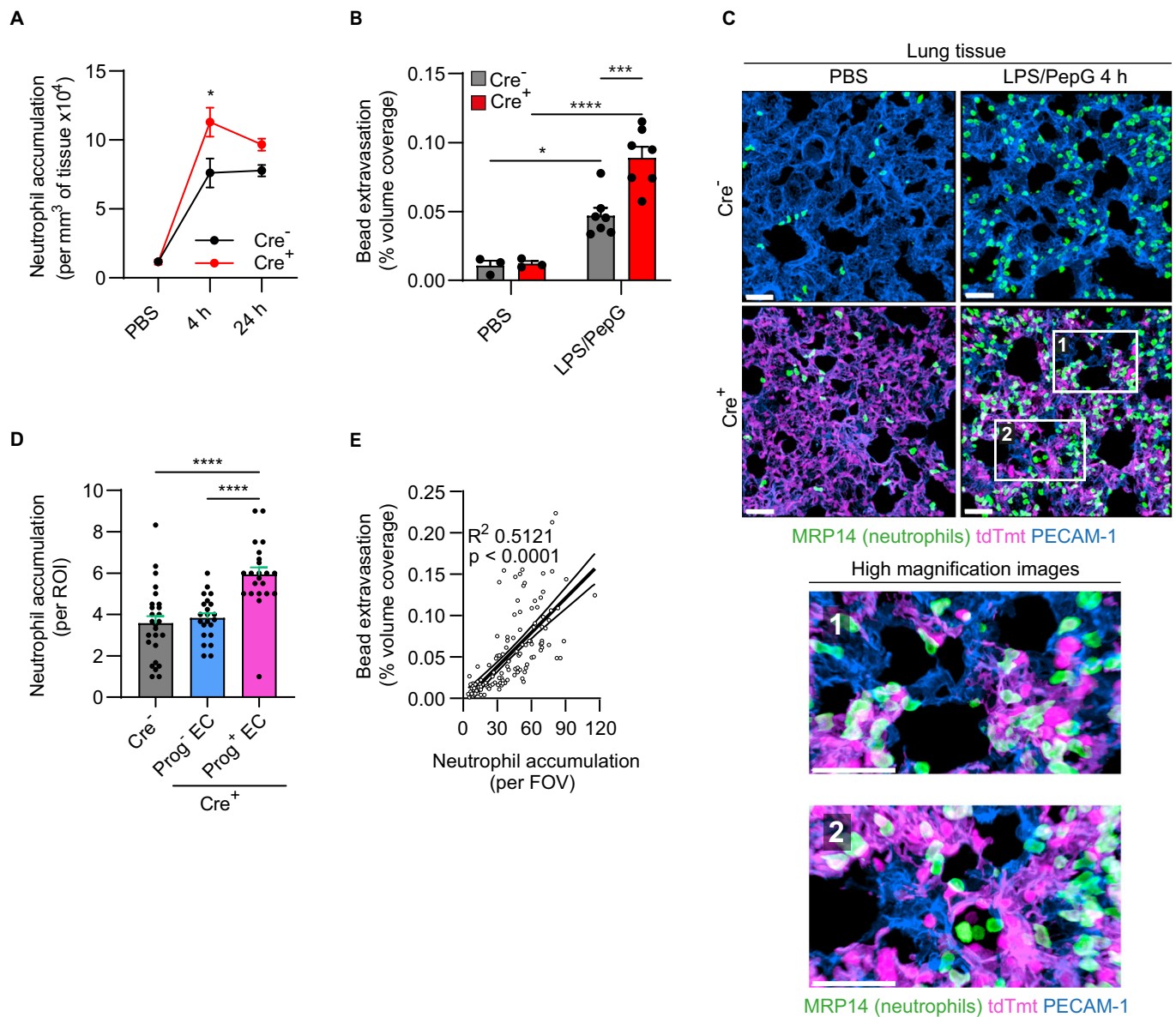

**Figure 5. Mice expressing EC progerin exhibit increased lung inflammation.**

Cre⁻ and Cre⁺ mice were subjected to systemic inflammation using a model of endotoxemia (LPS/PepG). (A) Neutrophil accumulation in lungs of Cre⁻ and Cre⁺ mice at the indicated times post-stimulation ($n = 3$–7 mice/group in 4 independent experiments). (B) Lung vascular leakage as quantified by extravasation of i.v. injected fluorescent microbeads ($n = 3$–7 mice/group in 4 independent experiments). (C) Representative confocal images of lung tissue sections of control and endotoxemic Cre⁻ and Cre⁺ animals immunostained for PECAM-1 (ECs) and MRP14 (neutrophils). White boxes delineate regions of interest, which are shown magnified in the lower panel. Scale bars: 30 μm. (D) Quantification of the number of neutrophils accumulating on lung ECs of Cre⁻ mice or on progerin negative versus positive lung ECs of the same Cre⁺ mice ($n = 3$–4 mice/group in 5 independent experiments). (E) Correlation between extravascular microbead accumulation and the number of neutrophils per field of view (FOV). The bold line shows linear regression with the 95% confidence intervals indicated by light lines ($n = 138$ images from 20 different mice in 5 independent experiments). Data information: (A, B, D, E) Data are mean ± SEM. *$P < 0.05$; ***$P < 0.001$; ****$P < 0.0001$. (A, B) Two-way ANOVA followed by Tukey's post hoc test. (D) One-way ANOVA followed by Tukey's post hoc test. (E) Pearson coefficient $R^2$ and $P$ value (deviation from zero slope). Source data are available online for this figure.

adhesion to IL-1β-stimulated untransduced (GFP⁻) and lamin A-transduced HUVECs was low, this response was significantly increased on progerin-transduced HUVECs (Fig. 8E,F). Furthermore, on adhesion to progerin-expressing cells, a greater number of neutrophils were ROS positive, as compared to neutrophils adherent to lamin A-transduced HUVECs (Fig. 8E,G). These data indicate an elevated activation state of neutrophils post adhesion to progerin-expressing ECs.

Together, our data with human cells provide additional evidence for progerin-expressing ECs being pro-senescent and presenting pro-inflammatory, pro-adhesive, and pro-stimulatory properties for neutrophils.

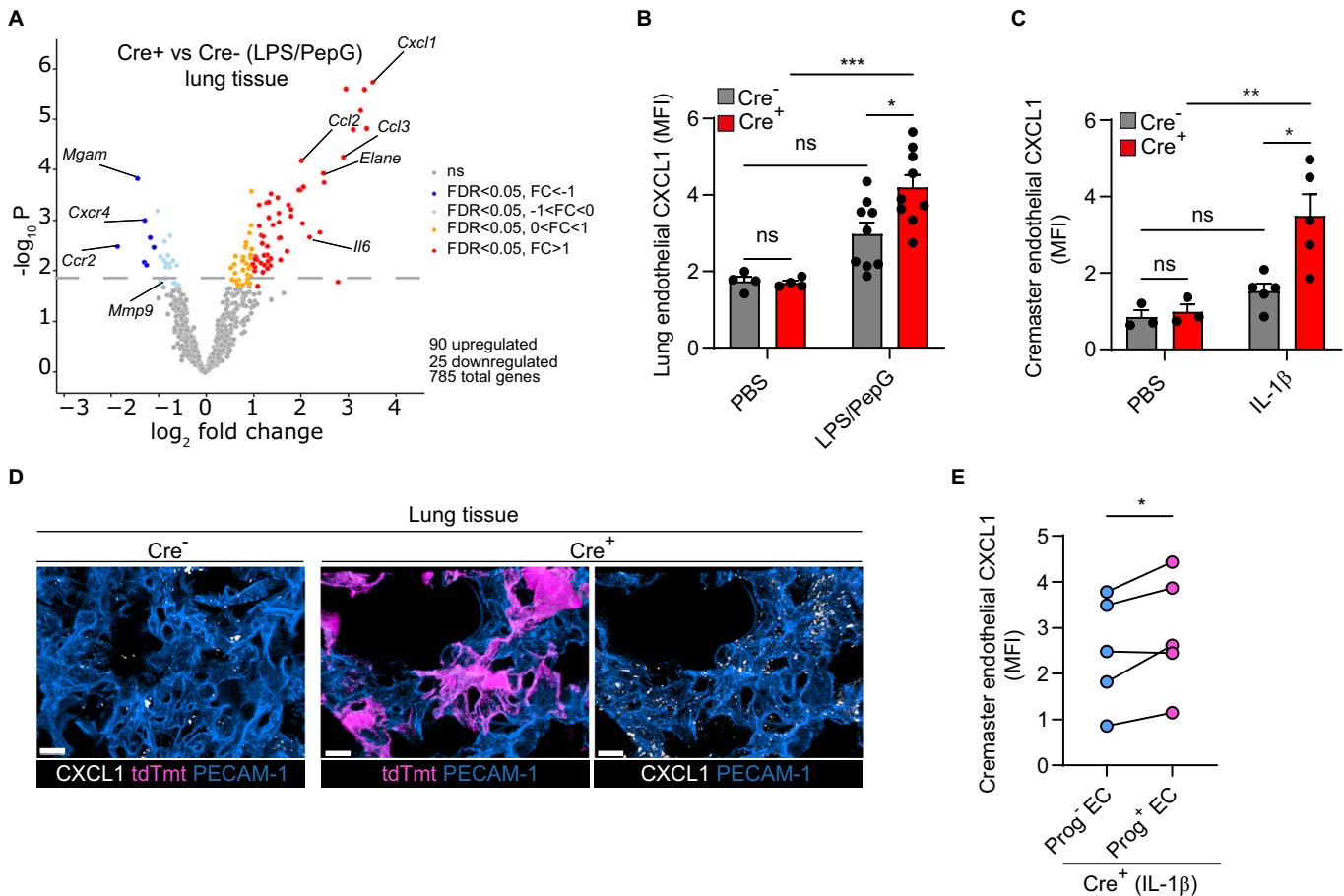

Figure 6. IL-1β-stimulated progerin-expressing endothelial cells exhibit a pro-inflammatory phenotype in vivo.

Cre⁻ and Cre⁺ mice were subjected to systemic inflammation using a model of endotoxemia (LPS/PepG, 2 h) or locally inflamed with intrascrotal injection of IL-1β. (A) Volcano plot of the relative difference in gene expression in whole lung tissue of stimulated Cre⁻ and Cre⁺ mice assayed by NanoString (785 genes; $n = 3$ mice/group in 1 experiment). Significantly differentially expressed genes (FDR > 0.05) in Cre⁺ lung tissue is color coded as follows: Downregulated genes in dark blue (fold change, FC < −1) and light blue (−1 < FC < 0); upregulated genes in orange (0 < FC < 1) and red (FC > 1). The presented results are based on Fisher's exact test with false discovery rate adjustment. See Methods for further analytical details. (B, C) Quantification of EC CXCL1 protein expression (MFI) in control and stimulated Cre⁻ and Cre⁺ mice as acquired by confocal microscopy in (B) lungs ($n = 4$–9 mice/group in 3 independent experiments) and (C) cremaster muscles ($n = 3$–5 mice/group in 2 independent experiments). (D) Representative confocal images of lung tissue sections of LPS/PepG-treated Cre⁻ and Cre⁺ animals immunostained for PECAM-1 (ECs) and CXCL1 with the signals exhibited in relation to progerin expressing ECs (tdTomato fluorescence). Scale bar: 30 μm. (E) CXCL1 expression (MFI) in progerin negative and positive IL-1β-stimulated cremasteric ECs ($n = 5$ mice/group in 2 independent experiments). Data information: (B, C) Data are mean ± SEM. *$P < 0.05$; **$P < 0.01$; ***$P < 0.001$; ns: not significant. Two-way ANOVA followed by Tukey's post hoc test. (E) *$P < 0.05$. Two-tailed paired Student's $t$-test. Source data are available online for this figure.

## Discussion

Dysregulated inflammation is a multifactorial feature of age-linked disorders. In investigating the associated mechanisms, altered immune cell functions have been extensively studied but the impact of aged ECs as the cause of uncontrolled inflammation remains poorly understood. As cellular senescence is closely aligned with molecular and functional deterioration of EC properties with age (Jia et al, 2019; Ting et al, 2021), here we investigated the impact of progerin-expressing ECs, used as a model of EC senescence, on neutrophil trafficking and activation. We report that senescent progerin-expressing ECs promote exaggerated neutrophil attachment to venules, leading to excessive neutrophil-dependent vascular leakage in models of inflammation. Mechanistically, we identified an elevated level of the chemokine CXCL1 on senescent ECs as driver of aberrant inflammatory responses. Together, our

findings shed light on the molecular basis of neutrophil-mediated enhanced vascular leakage in tissues harboring senescent ECs, a phenomenon that can contribute to the pathogenesis of multiple edematous ageing-linked and chronic inflammatory disorders.

Cellular senescence holds a prominent role in age-associated phenotypes and pathologies where it is aligned with both beneficial and detrimental effects (Muñoz-Espín and Serrano, 2014). Further, there is evidence of increased accumulation of senescent cells in various chronic inflammatory conditions such as in lungs in idiopathic pulmonary fibrosis (Justice et al, 2019; Schafer et al, 2017), in the aorta in atherosclerosis (Minamino et al, 2002; Roos et al, 2016), and in adipose tissue of patients with diabetes (Hickson et al, 2019; Tchkonia et al, 2013). These findings emphasize the need for better understanding of the expression and functional role of senescent cells in physiological and pathological settings. In this context, the use of senescence-ablator transgene mouse models

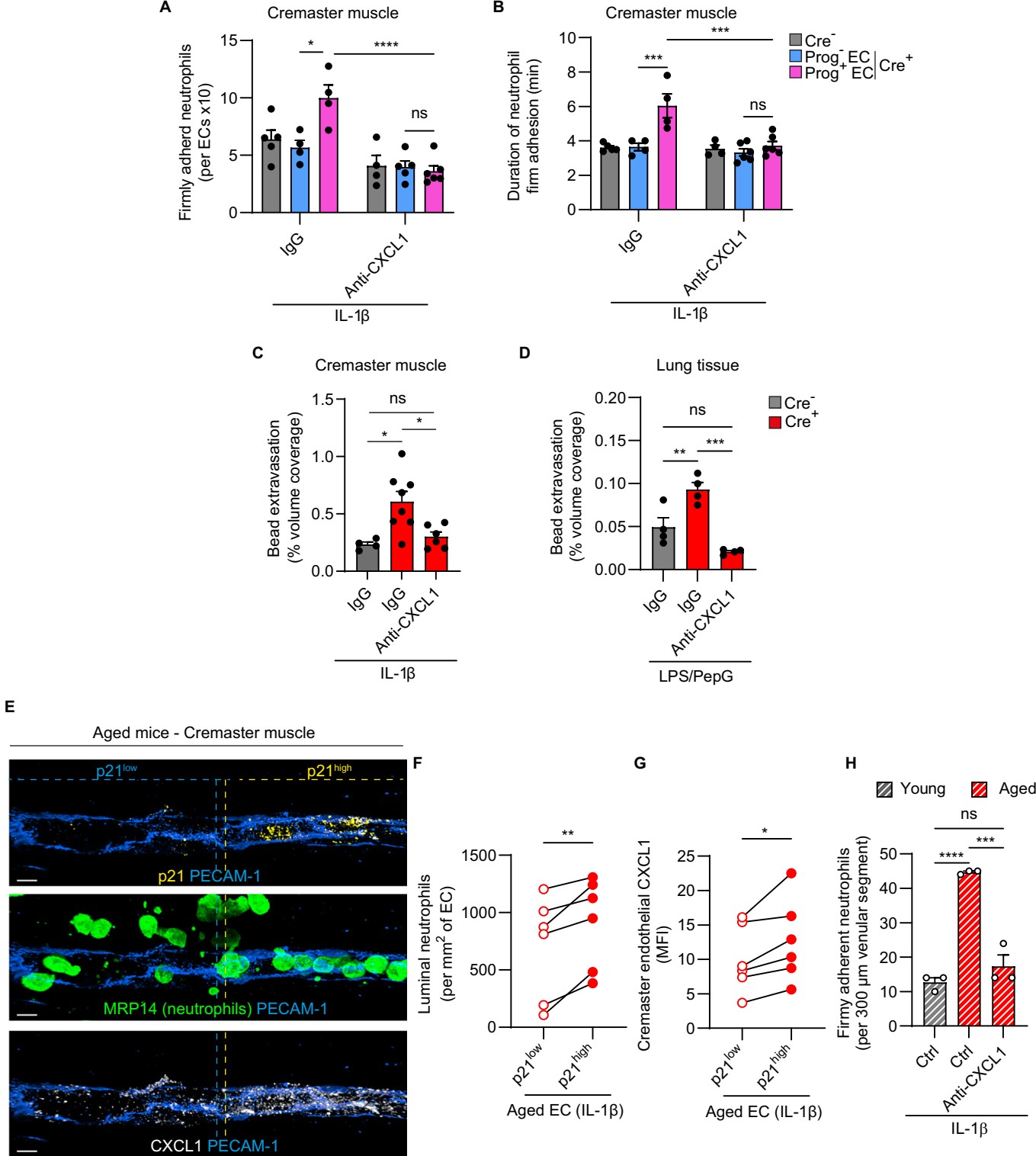

(Baker et al, 2016; Baker et al, 2011; Grosse et al, 2020), senolytic therapies (Camell et al, 2021; Hickson et al, 2019; Justice et al, 2019; Suda et al, 2021; Xu et al, 2018), and senescent reporter mice (Grosse et al, 2020), has shed much light on the localization and consequences of senescent cells in vivo. In addition, increased genetic and molecular knowledge of premature aging syndromes,

such as HGPS or progeria, has provided insight to fundamental cell biology of physiological aging (Carrero et al, 2016). Crucially, while sharing many molecular and symptomatic features with normal aging, HGPS is primarily a vasculopathy, characterized by accelerated vascular defects and early premature death due to severe cardiovascular conditions such as stroke and myocardial

**Figure 7. CXCL1 blockade in EC progerin expressing and aged mice normalizes dysregulated inflammatory responses.**

(A–C) Cremaster microvasculature of *Lyz2-EGFP-ki* Cre⁻ and Cre⁺ mice were acutely inflamed (IL-1β) and mice were treated with intravenously administered control (IgG) or neutralizing anti-CXCL1 mAbs. Neutrophil behaviors were analyzed by confocal intravital microscopy. (A) Number of firmly adherent neutrophils per progerin negative or positive EC ($n = 4$–6 in 19 independent experiments). (B) Duration of neutrophil firm adhesion to progerin positive versus negative ECs in the same vessel segment ($n = 4$–6 in 19 independent experiments). (C) Cremaster microvascular leakage as quantified by fluorescent bead extravasation ($n = 4$–8 mice/group in 5 independent experiments). (D) Lung microvascular leakage in mice treated with i.p. LPS/PepG and i.v. IgG or anti-CXCL1 Ab with bead extravasation expressed as percentage coverage ($n = 4$ mice/group in 2 independent experiments). (E–H) Cremaster muscles of aged mice were acutely inflamed with IL-1β. (E) Representative confocal images of cremaster muscles of IL-1β-treated aged animals immunostained for PECAM-1 (ECs), p21 (senescent EC), MRP14 (neutrophils) and CXCL1. Scale bar: 10 μm. (F) Number of luminal neutrophils per p21^low or p21^high ECs in the same venular segment ($n = 6$ mice/group in 2 independent experiments). (G) CXCL1 expression (MFI) in IL-1β-stimulated cremasteric ECs of aged mice ($n = 6$ mice/group in 2 independent experiments). (H) Aged *Lyz2-EGFP-ki* mice stimulated with IL-1β were treated with i.v. control (IgG) or neutralizing anti-CXCL1 mAbs, and the number of firmly adherent neutrophils was quantified ($n = 3$ mice/group in 9 independent experiments). Data information: (A–D, H) Data are mean ± SEM. *$P < 0.05$; **$P < 0.01$; ***$P < 0.001$; **** $P < 0.0001$; ns: not significant. (A, B) Two-way ANOVA followed by Tukey's post hoc test. (C, D, H) One-way ANOVA followed by Tukey's post hoc test. (F, G) Two-tailed paired Student's *t*-test. Source data are available online for this figure.

infarction (Gordon et al, 2014). In line with this, progeroid mouse models have proven to be of special value in investigations of cardiovascular disorders such as atherosclerosis (Hamczyk et al, 2019; Marcos-Ramiro et al, 2021; Sánchez-López et al, 2021), vascular fibrosis (Osmanagic-Myers et al, 2018), and metabolic diseases (Barinda et al, 2020; Kreienkamp et al, 2019). Here, we used a mouse line carrying a progeroid mutation in ECs as a reductionist model of vascular aging. Of significance, by combining this conditional mouse model with the Rosa-26-Tomato Cre reporter, we established an in vivo tool in which progerin-expressing and non-expressing ECs could be distinguished in the same tissue and often within the same venular segment. This powerful feature provided direct evidence of progerin-expressing ECs presenting a senescence-associated phenotype in vivo, findings that are in agreement with previously published works (Sun et al, 2020). Functionally, mice expressing EC progerin supported excessive neutrophil-EC adhesion, responses that are highly reminiscent of those in inflamed microcirculation of physiologically aged mice (Barkaway et al, 2021). Indeed, as noted with progerin-expressing ECs, stimulated aged ECs exhibiting a senescent phenotype supported a greater level of neutrophil attachment. Together, the results identify senescent ECs as an instigator of aberrant neutrophil behavior that could be an integral component of the pro-inflammatory state of aged tissues. Interestingly, this phenomenon appeared to be mediated in a cell-autonomous manner, a response that was most dramatically illustrated in real-time tracking of neutrophils by confocal IVM. Hence, although paracrine effects of senescent cells on non-senescent cells, as elicited by SASP secreted factors, have been reported in multiple settings, most notably in models of tumor growth (Muñoz-Espín and Serrano, 2014), our results suggest a profound resilience of non-senescent ECs to SASP. Supported by previous findings stemming from cultured EC systems (Barinda et al, 2020), our observations are in line with the critical role of ECs in regulating physiological vascular responses and maintaining tissue home-ostasis. To our knowledge, the present study is the first to report on excessive neutrophil attachment to stimulated senescent micro-vascular ECs in real-time in vivo, identifying a previously unreported mechanism of ageing-associated pathological vascular damage.

The complex interplay between senescent ECs and neutrophil responses has also been illustrated in other in vivo settings. For example, using a mouse model of ischemic retinopathy, senescent vascular units were found to activate neutrophils to release NETs, a process that led to clearance of senescent ECs and reparative

vascular regeneration (Binet et al, 2020). In contrast, while not directly investigating the involvement of vascular cells, activated ROS-generating neutrophils have been aligned with induction of cellular senescence in a murine model of acute liver injury and in livers of aged mice (Lagnado et al, 2021). In agreement with our findings, the latter study provided evidence for the ability of senescent cells to promote neutrophil infiltration in aged livers. Of relevance, we have previously shown that compromised EC autophagy, a hallmark of aging (Schmauck-Medina et al, 2022), induces excessive neutrophil infiltration in multiple models of acute inflammation (Reglero-Real et al, 2021). Collectively, while there is evidence for both physiological and pathological roles of cellular senescence (Muñoz-Espín and Serrano, 2014), our findings provide direct evidence for senescent ECs driving dysregulated immune cell trafficking.

Infiltration of neutrophils into tissues is exquisitely regulated by EC-expressed adhesion and chemotactic molecules, events that are subject to temporal and spatial regulation (Girbl et al, 2018; Nourshargh and Alon, 2014). Indeed, we have previously shown that the ELR⁺ CXC chemokines CXCL1 and CXCL2 act sequentially to guide neutrophils through stimulated venular walls, as governed by their distinct cellular source and presentation to migrating immune cells (Girbl et al, 2018). Within this cascade, EC-derived CXCL1 was identified as a key regulator of neutrophil firm arrest and crawling on cytokine-stimulated venular walls. In investigating the mechanism of increased neutrophil attachment to progerin-expressing venules, targeted transcriptomic analysis and immunofluorescence staining identified enhanced levels of the chemokine CXCL1 on stimulated progerin-expressing ECs in vivo. Crucially, functional blockade of CXCL1 abrogated increased attachment to and crawling of neutrophils on inflamed progerin-expressing venular segments and neutrophil adhesion to stimulated senescent aged ECs. In line with our findings, it has previously been shown that blockade of CXCR2, the principal neutrophil cognate receptor for CXCL1, suppresses neutrophil infiltration and exerts protection against lung inflammation in aged mice (Nomellini et al, 2012; Nomellini et al, 2008). Together, these results confirm the efficacy of CXCL1 as a key regulator of neutrophil-EC interactions in inflamed aged tissues and indeed there is ample evidence of dysregulated and pathological neutrophil migratory behaviors in different inflammatory settings following disrupted CXCL1 gradient (Barkaway et al, 2021; Girbl et al, 2018; Owen-Woods et al, 2020). In addition, progerin-expressing cells exhibited elevated mRNA or protein levels of key EC adhesion molecules in different settings, and indeed showed an upregulation of numerous pro-

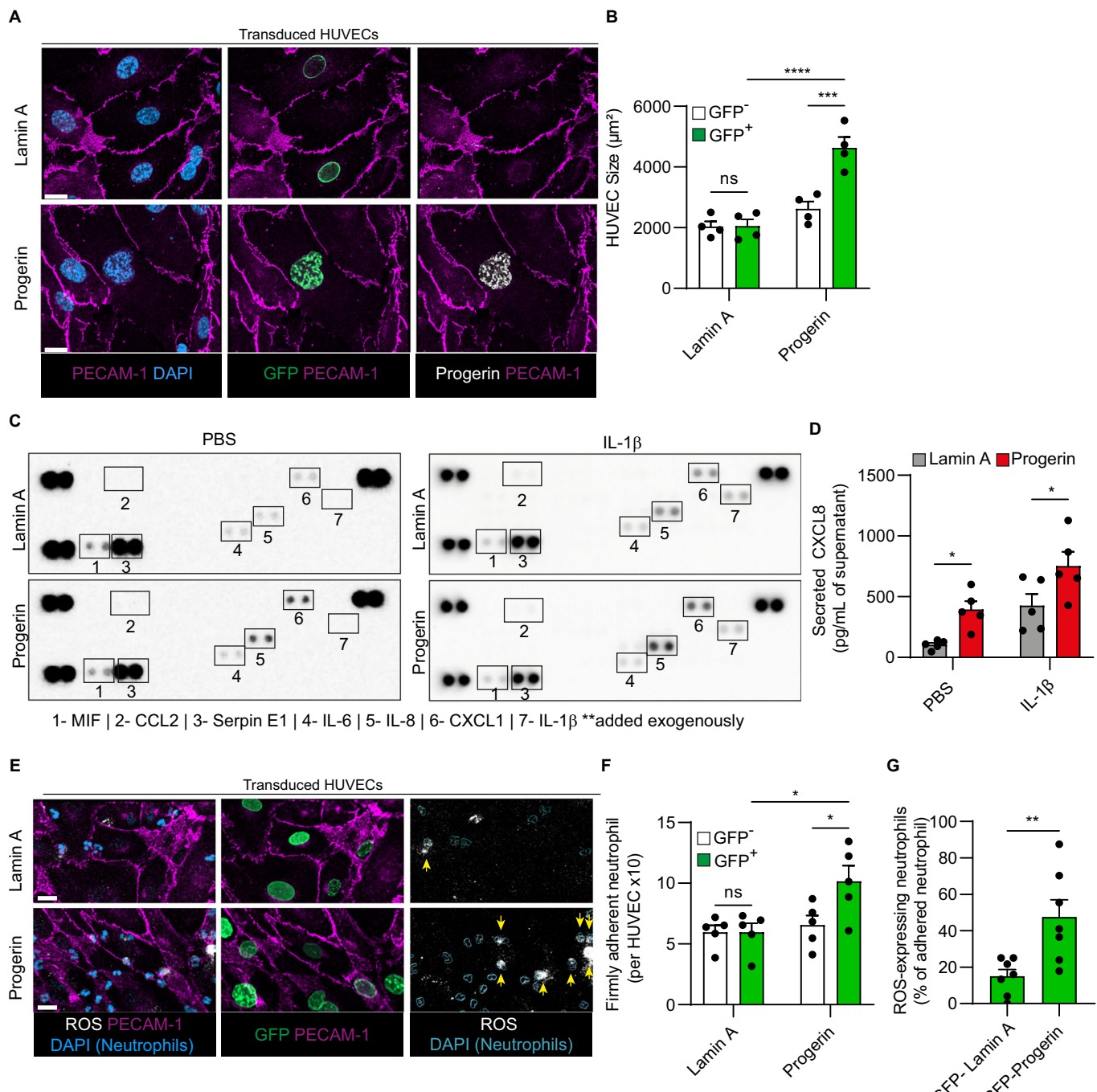

1- MIF | 2- CCL2 | 3- Serpin E1 | 4- IL-6 | 5- IL-8 | 6- CXCL1 | 7- IL-1β **added exogenously

inflammatory genes. This apparent overall primed state of senescent ECs could support faster and more excessive immune cell infiltration. This phenomenon was indeed noted in our pathological lung injury model and may be more pronounced in chronic inflammatory settings.

In addition to mediating the passage of immune cells across venular walls, ECs play a critical role in regulating vascular permeability to macromolecules. As with aberrant immune cell migration, excessive vascular permeability is a feature of numerous inflammatory pathologies (Claesson-Welsh et al, 2021; Oakley, 2014). Enhanced vascular leakage can be elicited by mediators that

act directly on the endothelium, such as VEGF and histamine, or by vasoactive factors released by neutrophils when in close apposition to venular ECs (Distasi and Ley, 2009). In physiological inflammation, the latter likely include neutrophil-secreted mediators such as TNF, PAF, and neutrophil-derived VEGF (Distasi and Ley, 2009; Finsterbusch et al, 2014; Massena et al, 2015; Tecchio et al, 2014), while in pathological settings, neutrophil-derived ROS, proteases, and/or NETs may promote vascular permeability via EC injury (Distasi and Ley, 2009). Here we show that in acute inflammatory settings, the sustained attachment of neutrophils to senescent ECs leads to neutrophil-dependent microvascular

**Figure 8. Progerin-expressing human ECs support increased neutrophil adhesion and activation.**

(A) Representative confocal images of GFP-lamin A- or GFP-progerin-transduced HUVECs stained for PECAM-1, progerin, and DAPI. Scale bar: 5 μm. (B) Cell size of lamin A-GFP and progerin-GFP transduced HUVECs (GFP⁻: non-transduced) ($n = 4$ in 2 independent experiments). (C) Analysis of secreted proteins from PBS (control) and IL-1β-stimulated GFP-lamin A and GFP-progerin-expressing HUVECs, as assayed using a multiplex antibody array (targeting 36 proteins). (D) CXCL8 levels in control (PBS) and IL-1β stimulated transduced HUVECs ($n = 5$ transduced cell cultures per group, analyzed by ELISA in 4 independent experiments). (E–G) Lamin A-GFP- and progerin-GFP-transduced HUVECs were stimulated with IL-1β and co-incubated with purified human neutrophils loaded with a ROS tracker. (E) Representative image of washed and fixed cells stained for PECAM-1, DAPI and ROS. The DAPI signal intensity for HUVEC nuclei was thresholded out using IMARIS software based on their dimmer fluorescence compared to neutrophil nuclei. Yellow arrows highlight ROS-positive neutrophils. Scale bars: 20 μm. (F) Number of neutrophils firmly adherent to non-transduced (GFP⁻) and lamin A- or progerin-transduced (GFP⁺) HUVECs as quantified per ten cells for each data point (mean ± SEM, $n = 5$ in 5 independent experiments). (G) Number of ROS-positive neutrophils adherent to GFP-lamin A- or GFP-progerin-transduced HUVECs, expressed as a percentage of the total number of neutrophils adherent to the respective HUVEC transductants ($n = 7$ in 7 independent experiments). Data information: (B, D, F, G) Data are mean ± SEM. *$P < 0.05$; **$P < 0.01$; ***$P < 0.001$; **** $P < 0.0001$; ns: not significant. (B, D, F) Two-way ANOVA followed by Tukey's post hoc test. (G) Two-tailed paired Student's $t$-test. Source data are available online for this figure.

leakage. Indeed, blockade of neutrophil adhesion to venular ECs by an anti-CXCL1 mAb totally abrogated the increased vascular leak quantified in EC progerin-expressing mice. Of note, we detected no difference in the efficacy of histamine, a potent direct-acting vascular permeability inducing factor, to elicit vascular leakage in mice expressing EC-progerin as compared to controls. Collectively, these results suggest that in the acute inflammatory settings analyzed, compromised barrier function of the endothelium is directly aligned with aberrant stimulation of ECs by adherent neutrophils. In addressing, and specifically hypothesizing that increased attachment of neutrophils to senescent ECs may lead to excessive neutrophil activation with subsequent release of EC permeability factors, we detected increased release of ROS from neutrophils adherent to progerin-expressing HUVECs. As neutrophil-derived ROS can induce compromised vascular integrity in numerous inflammatory settings (Mittal et al, 2014), the present results suggest that premature activation of the neutrophil NADPH oxidase within the lumen of aged and/or senescent blood vessels may be a significant driver of neutrophil-mediated vascular injury and permeability in ageing-linked and chronic cardiovascular pathologies. Of note, since stimulated progerin-expressing HUVECs exhibited an enhanced secretory profile, the mechanism of increased permeability in blood vessels harboring senescent ECs may involve multiple mediators. This could include factors stemming from both activated neutrophils and ECs and hence requires further exploration.

Together, we provide conclusive evidence for senescent ECs driving excessive neutrophil trafficking and neutrophil-dependent vascular leakage. The findings suggest that greater understanding of altered functionality of senescent ECs in vivo could identify therapeutic targets aimed at suppressing ageing-linked inflammation.

## Methods

### Reagents and tools table

| Reagent/Resource | Reference or source | Identifier or catalog number |
|---|---|---|
| **Experimental models** | | |
| *Lmna^LCS/LCS* | Osorio et al, 2011 | |
| *B6;129S6. Gt(ROSA) 26Sor^tm9(CAG-tdTomato)Hze/J* | Jackson lab | #007905 |

| Reagent/Resource | Reference or source | Identifier or catalog number |
|---|---|---|
| *Tie2-Cre;CAG-CAT-Z double-transgenic mice* | Kisanuki et al, 2001 | |
| *Lyz2-EGFP-ki* | Dr Thomas Graf (Center for Genomic Regulation and ICREA, Spain) | |
| >18-month-old aged mice | Janvier labs | Cat #SC-C57J-M |
| Mouse lung endothelial cells | This study | See Methods section |
| Human umbilical vein endothelial cells (C-pooled) | Promocell | Cat #C12203 |
| Human neutrophils | This study | See Methods section |
| **Recombinant DNA** | | |
| Neo GFP-Progerin | Addgene | Cat #118710 |
| pLenti CMV TRE3G Neo GFP-Lamin A wild type | Addgene | Cat #118709 |
| pLenti CMV rtTA3 Hygro (w785-1) | Addgene | Cat #26730 |
| **Antibodies** | | |
| Anti-mouse progerin | This study | See Methods section |
| Anti-mouse PECAM-1 clone 390 | Thermo Fisher Scientific | Cat #16-0311-85 |
| Anti-mouse VE-Cadherin Clone eBioBV14 | Thermo Fisher Scientific | Cat #14-1442-85 |
| Anti-mouse ICAM-1 Clone YN1/1.7.4 | Thermo Fisher Scientific | Cat #14-0541-85 |
| Anti-mouse ICAM-2 Clone 3C4 | Biolegend | Cat #105602 |
| Anti-mouse VCAM-1 Clone 429 | Biolegend | Cat #105702 |
| Anti-mouse CD29 Clone HMβ1-1 | Biolegend | Cat #102202 |
| Anti-mouse CD45 Clone 30-F11 | Biolegend | Cat #103104 |
| Anti-mouse CD115 Clone AFS98 | Thermo Fisher Scientific | Cat #14-1152-85 |

| Reagent/Resource | Reference or source | Identifier or catalog number |
|---|---|---|
| Anti-γH2AX Ser139 Clone 20E3 | Cell Sigaling technology | Cat #9719S |
| Anti-mouse CXCL1 (for tissue staining) | R&D Systems | Cat #AF-453-NA |
| Anti-mouse CXCL1 (blocking experiments) clone 48415 | R&D Systems | Cat #MAB453-100 |
| Anti-mouse p21 Clone EPR18021 | Abcam | Cat #ab237264 |
| Anti-human PECAM-1 Clone WM59 | Thermo Fisher Scientific | Cat #14-0319-82 |
| Anti-β-Actin Clone AC-15 | Sigma-Aldrich | Cat #A5441-100UL |
| Anti-GFP Clone 3H9 | ChromoTek | Cat #ABIN398304 |
| Anti-Histone H2B Clone W17025A | Biolegend | Cat #606301 |
| Anti-MRP14 | Gift from Dr Nancy Hogg (Francis Crick Institute, UK) | |
| Anti-mouse progerin | This study | See Methods section |
| **Oligonucleotides and sequence-based reagents** | | |
| PCR primers | This study | See Methods section |
| **Software** | | |
| Imaris 9 | Oxford Instruments | |
| FlowJo 10 | Flowjo, LLC | |
| **Other** | | |
| Zeiss LSM800 microscope | Zeiss | Bio-Imaging Facility \| CMR (centre-for-microvascular-research.com) |
| Leica SP5 microscope | Leica | Bio-Imaging Facility \| CMR (centre-for-microvascular-research.com) |
| Leica SP8 microscope | Leica | Bio-Imaging Facility \| CMR (centre-for-microvascular-research.com) |

## Animals

$Lmna^{LCS/LCS}$ mice (8–12 weeks old) used for all studies were generated as previously described (Osorio et al, 2011). To achieve progerin expression in endothelial cells, $Lmna^{LCS/LCS}$ were intercrossed with the Tie2-Cre transgenic line kindly provided by Prof. Triantafyllos Chavakis (Kisanuki et al, 2001) and subsequently intercrossed with $B6;129S6. Gt(ROSA)26Sor^{tm9(CAG-tdTomato)Hze}/J$ (JAX-Lab stick #007905) (Madisen et al, 2010) giving rise to the $Tie2-Cre;Lmna^{LCS/LCS};Rosa26^{tdTomato/tdTomato}$ mice termed "Lmna mice" in this paper. Commonly used at 8–12 weeks old, Lmna mice exhibited no abnormal health conditions throughout the entire duration of the study and up to the age of 6 months. To enable visualization of neutrophil behaviors by intravital microscopy,

Lmna mice were intercrossed with the neutrophil reporter $Lyz2$-$EGFP$-$ki$ mouse line. This strain was kindly provided by Dr Markus Sperandio (Ludwig Maximilians University of Munich, Germany) and used with the permission of Dr Thomas Graf (Center for Genomic Regulation and ICREA, Spain). $Lyz2$-$EGFP$-$ki$ mice have EGFP cDNA knocked into the lysozyme M ($Lyz2$) locus to generate EGFP$^+$ myeloid cells (GFP$^{bright}$ neutrophils, GFP$^{dim}$ monocytes and macrophages) (Faust et al, 2000). The compound $Tie2-Cre;Lmna^{LCS/LCS};Rosa26^{tdTomato/tdTomato}; Lyz2$-$EGFP$-$ki$ mutant is here referred to as $Lmna$ mice;$Lyz2$-$EGFP$-$ki$. Aged C57BL/6JRj mice (Janvier Labs, Le Genest-Saint-Isle, France) were used at ≥18+ months old. All animals were group housed in individually ventilated cages under specific pathogen-free (SPF) conditions and a 12 h light-dark cycle. Animals were humanely sacrificed via cervical dislocation at the end of experiments in accordance with UK Home Office regulations. For the study, male and female mice were employed, with male specimens specifically chosen to collect cremaster muscle samples. Unless constrained by evident phenotypic alterations (e.g., leukocyte recruitment solely observed in stimulated animals but not in control subjects), all experiments involved steps in which experimenters were blinded.

## Antibodies

The following Abs were obtained commercially: anti-mouse PECAM-1 (Clone 390, Thermo Fisher Scientific), anti-mouse VE-Cadherin (clone eBioBV14, Thermo Fisher Scientific), anti-mouse ICAM-1 (clone YN1/1.7.4, Thermo Fisher Scientific), anti-mouse ICAM-2 (clone 3C4, Biolegend), anti-mouse VCAM-1 (clone 429; MVCAMA.A, Biolegend), anti-mouse CD29 (clone HMβ1-1, Biolegend), anti-mouse CD45 (clone 30-F11, Biolegend), anti-mouse CD115 (AFS98, Thermo Fisher Scientific), anti-γH2AX Ser139 (clone 20E3, Cell Signaling Technology), anti-mouse CXCL1 for staining (goat polyclonal Ab, R&D Systems), anti-mouse CXCL1 for blocking experiments (clone 48415, R&D Systems), anti-mouse p21 (Clone EPR18021, Abcam) anti-human PECAM-1 (clone WM59, Thermo Fisher Scientific), anti-β-Actin (clone AC-15, Sigma-Aldrich), anti-GFP (clone 3H9, ChromoTek) and anti-Histone H2B (W17025A, Biolegend). Anti-MRP14 was a gift from Dr Nancy Hogg (Francis Crick Institute, London, UK).

### Anti-mouse progerin antibody

To generate a specific antibody to mouse progerin, standard rabbit immunization protocols (Covalab, France) were employed using a peptide corresponding to amino acids 603 to 610 unique to the mouse progerin sequence (GAQSSQNC). Briefly, two rabbits were injected with a priming peptide (KAAGGAGAQSSQNCSIM) 3 times over the course of 53 days. Subsequently, two booster injections of a shorter peptide sequence (AGAQSSQNCS) were administered over 30 days, before the final bleed on day 137. The polyclonal antiserum was affinity purified and its specificity for progerin validated by immunofluorescence staining and immunoblotting.

## Plasmids

pLenti CMV TRE3G Neo GFP-Progerin was a gift from Tom Misteli (Addgene plasmid # 118710; RRID:Addgene_118710) (Kubben et al, 2016); pLenti CMV TRE3G Neo GFP-Lamin A wild type was a gift from Tom Misteli (Addgene plasmid # 118709; RRID:Addgene_118709) (Kubben et al, 2016); pLenti CMV rtTA3

Hygro (w785-1) was a gift from Eric Campeau (Addgene plasmid # 26730; RRID:Addgene_26730).

## Confocal intravital microscopy of the mouse cremaster muscle

Anesthetized (3–5% isoflurane) male mice received an intrascrotal (i.s.) injection of fluorescently labeled anti-PECAM-1 mAb (4 µg) and/or IL-1β (50 ng, R&D Systems) to label blood vessels within the tissue and to induce an inflammatory response, respectively. Mice were anaesthetized by i.p. injection of ketamine (100 mg/kg) and xylazine (10 mg/kg). Cremaster muscles were surgically exteriorized for intravital imaging 2 h after IL-1β administration as described previously (Colom et al, 2015; Woodfin et al, 2011). In some experiments, anti-CXCL1 mAb or isotype control IgG2a (1 mg/kg) was injected i.v. via cannulation as indicated in the relevant sections and figure legends. Postcapillary venules (20–45 µm diameter, identified by their morphology and direction of blood flow) of flat mounted cremaster muscles were imaged every 30 to 60 s for 0.5 to 2.0 h using an upright Leica SP5 or SP8 laser scanning confocal microscope equipped with a 20x/1.0 water-dipping objective lens and 8 kHz resonant scanner. Resultant time-lapse z stack images were assembled offline into 3D videos using IMARIS software (Oxford Instruments, ANDOR). The x × y × z dimensions of the recorded area were typically $300 \times 130 \times 35$ µm and the resulting voxel size $0.29 \times 0.29 \times 0.69$ µm.

### Neutrophil migration mode and dynamics

Neutrophil speed and displacement were determined by manual tracking of individual neutrophils using IMARIS software and migratory paths were graphically illustrated using the Prism 9 software (GraphPad). Luminal neutrophils were defined as adherent when they remained stationary on the endothelium for at least 30 s and as intraluminal crawling was defined by displacement of neutrophils by at least 2 cell diameters on the endothelium over the entire observation period. The number of adherent and crawling neutrophils was quantified by the mean number of neutrophils counted at 4 time points per mouse.

## Quantification of microvascular leakage

Quantification of vascular leakage in the cremaster muscle or lung (also refer to fix tissue preparation protocol below) was performed using fluorescent microbeads (Le et al, 2015). Briefly, post i.p. injection of PBS-LPS/PepG or intrascrotal injection of PBS or IL-1β, mice were injected i.v. with Crimson-(625/645)-microbeads (20 nm in diameter, $3.6 \times 10^{12}$ microbeads/g of mouse; Thermo Fisher Scientific) between 0.5 to 2 h before sacrifice and vascular permeability was quantified in lungs and cremaster muscles, respectively. Microbead leakage into the extravascular space was defined as microbead coverage and quantified as a percent of the total volume occupied by the microbeads over the total extra-vascular volume of the section for lung, or extravascular volume within 20 µm perimeter outside of the venule for cremaster muscle.

## Mile's assay

Vascular leakage was assessed using the Mile's assay as previously described (Colom et al, 2012). Briefly, mice were anaesthetized by

i.p. injection of ketamine (100 mg/kg) and xylazine (10 mg/kg). Evans blue dye (1% in PBS, 150 µL per mouse) was injected i.v. After 15 mins, LTB$_4$ (100 ng for 1 h; Cayman Chemicals), histamine (300 ng for 30 min; Sigma) or PBS (control) were administered in 20 µL volumes intradermally into the dorsal skin or intrascrotally (histamine, 4 µg in 400 µl for 30 min). At the end of the experiment, animals were sacrificed, and stimulated skin sites were harvested, weighted, and incubated in formamide (Sigma) at 56 °C overnight. The amount of extracted Evans blue was quantified by spectroscopy at 620 nm.

## Immunofluorescence staining and confocal analysis of fixed tissues

### Cremaster muscle and ear preparation

Whenever relevant, tissues were immunostained with injected fluorescently labeled anti-PECAM-1 antibody in situ for 2–4 h. Control and IL-1β stimulated tissues were surgically removed and fixed in ice-cold 4% paraformaldehyde (PFA) in PBS for 1 h, blocked/permeabilized at room temperature for 5 h in PBS containing 25% FCS and 0.5% Triton X-100, and incubated overnight at 4 °C with primary antibodies to one or more of the following proteins: progerin, γH2AX, VE-Cadherin, CXCL1 followed by staining with appropriate Alexa-Fluor conjugated secondary antibodies for 3 h at room temperature when necessary. Whole tissues were mounted with PBS under glass cover slips prior to confocal microscopy.

### Lung preparation

Immunostaining of sectioned lungs was performed on inflamed or naïve control mice. Lungs of sacrificed mice were perfusion-fixed with 2% PFA in PBS for ~2 min, then inflation-fixed for ~5 min with 4% PFA/OCT (1:1) and finally immersed in 4% PFA for 2 h. Fixed lungs were then successively immersed into 10%, 15%, and 30% sucrose for 2 h, overnight and 4 h, respectively, and frozen in optimal cutting temperature medium (OCT; Thermo Fisher Scientific). 30-µm-thick frozen sections were mounted onto SuperFrost microscope slides, and tissues blocked/permeabilized at room temperature for 30 min with 10% FCS and 0.5% Triton X-100 (Sigma-Aldrich) in PBS. Immunostaining was performed by incubating tissues for 1 h at room temperature with primary antibodies and subsequently with secondary antibodies if required. Finally, immunostained tissues were mounted with ProLong gold mounting medium (Thermo Fisher) under glass cover slips.

### Confocal imaging of immunostained tissues and quantifications

Immunostained tissues were imaged with an inverted Zeiss LSM800 laser-scanning confocal microscope. To capture serial optical z sections, images were acquired with 20x/0.8 air, 40x/1.3, oil or 63x/1.4 oil objective lenses, with a 0.07–0.62 µm pixel size (in XY plane) and 0.23–1.0 µm voxel depth. Endothelial cells (ECs) were identified as ICAM-2, VE-Cadherin or PECAM-1 positive, and neutrophils as MRP-14 or EGFP positive cells. Capillaries distinction was made based on their different diameters (<10 µm for capillaries). Postcapillary venules were distinguished from arterioles in cremaster muscles and ears by the distinctive morphology of their respective ECs (more elongated in arterioles), vessel straightness, and interactions with neutrophils, which exclusively occur in venules of ears and cremaster muscles.

To determine the association of neutrophils with progerin expressing tdTomato$^+$ microvascular segments, lung section images were segmented in 2 to 4 tdTomato$^-$ and tdTomato$^+$ ROIs identical in size, with IMARIS software. Neutrophil counts within each ROI were determined and reported as the average number of neutrophils per ROI per image or field of view. The expression of molecules associated with EC structures (i.e., γH2AX and CXCL1) was quantified in 3D reconstructed images with IMARIS software and expressed as mean fluorescence intensity (MFI) for CXCL1 or the number of fluorescent Foci (diameter >0.7 μm) for γH2AX.

### Neutrophil depletion

Neutrophil depletion was performed using a technique developed and previously published by our group to selectively deplete neutrophils without affecting monocyte numbers (Voisin et al, 2009). Briefly, mice were injected intraperitoneally (i.p.) with 25 μg/mouse/day of anti-GR1 antibody (RB6-8C5) for 3 consecutive days preceding the induction of the inflammatory response. Neutrophil depletion (~99.7%), with no effect on monocyte numbers, was validated in tail vein blood samples by flow cytometry as described below.

### Lung cell dissociation for proteomic and transcriptomic analysis

Lungs from control and LPS/PepG stimulated mice were flushed with 10 mL of wash buffer (2 mM EDTA, 20 U/mL Heparin and 0.5% BSA in PBS) via the right ventricle to remove red blood cells and leukocytes from the vascular lumen. Lungs were subsequently dissected out, minced with crossed scalpels and digested in Type-1 collagenase (500 U/mL, Gibco) and DNase (200 U/mL, Sigma) in PBS with calcium and magnesium for 1 h at 37 °C. After incubation in ACK erythrocyte lysis buffer (150 mM $NH_3Cl$, 1 mM $KHCO_3$, and 1 mM EDTA), cells were pelleted by centrifugation and used for subsequent flow cytometry, RT-qPCR and NanoString applications.

### Analysis and sorting of lung samples by flow cytometry

Phenotyping of lung endothelial cells (progerin expression, SA-β-Gal activity, morphology), endothelial cell sorting from lungs, and neutrophil counts from whole blood were assessed by flow cytometry. Lung cells were dissociated as described above; whole blood was collected in PBS containing 50 mM EDTA via the inferior vena cava or 5 mM EDTA via tail vein bleed. When necessary, a red blood cell lysis was performed by incubating samples in ACK erythrocyte lysis buffer for 3 to 5 min. Samples were then washed and incubated in staining buffer (2 mM EDTA and 0.5% BSA in PBS) containing anti-CD16/CD32 mAbs to block FC receptors and relevant fluorophore-conjugated primary antibodies at 4 °C for 30 min. Using a LSR Fortessa II (BD Biosciences) flow cytometry analyzer, doublet, and dead cells were excluded and gating of cells of interest was performed as follows: CD45$^+$ Ly6G$^+$ CD115$^-$ (neutrophils) and CD45$^-$ PECAM-1$^+$ ICAM-2$^+$ tdTomato$^{-/+}$ (ECs). Accurate cell counts were validated using 123count eBeads Counting Beads (ThermoFisher Scientific). TdTomato-positive and negative endothelial cells were

sorted using an Aria II or Aria Fusion (BD Biosciences) fitted with a 100 μm nozzle. FlowJo software (TreeStar) was used for data quantification.

### NanoString analysis

Total RNA was extracted from whole lung cells of control or IL-1β stimulated mice using RNeasy Mini kit (QIAGEN) and stored at −80 °C. Isolated RNA samples were subjected to quality and quantity control via assessment of the RNA integrity number (RIN) by Agilent 4200 TapeStation (Agilent Technologies, UK) and RNA concentration using a Qubit 4.0 fluorometer (Thermo Fisher Scientific, UK). Gene expression profiles were assessed using the NanoString nCounter SPRINT Analysis System and the Host Response Panel according to manufacturer's protocol (NanoString Technologies, USA). Raw data were analyzed using nSolver 4.0 Software (NanoString Technologies, USA) and were normalized against the housekeeping genes selected using GeNorm software. Raw counts of 785 genes extracted from NanoString Host Pathogens panels were used for Differential Gene Expression (DGE) analysis. This was based on negative binomial distribution using regression models of the normalized count data via DESeq2 (v1.37.4) and a Wald test to compare variation between PBS and LPS/PepG and, LPS/PepG Cre$^-$ and Cre$^+$ groups in lung tissue samples. $P$ values were false discovery rate (FDR) adjusted using Storey's q values with a cut-off of q < 0.05 used to determine significance in differentially expressed genes (DEGs). The distribution of DEGs was illustrated in a volcano plot using the easylabel (v0.2.4) R package. DEGs that demonstrated upregulation in the Cre$^+$ group were used for over-representation analysis via the PANTHER online tool (20220202 release). The 81 upregulated genes were tested against the whole reference gene panel (785 genes) to determine whether a particular Gene Ontology (GO) biological process (GO database https://doi.org/10.5281/zenodo.6399963, released 2022-03-22) was overrepresented. The presented results are based on Fisher's exact test with false discovery rate adjustment.

### Real-time RT-qPCR analysis of lung tissue and ECs

Total RNA was purified from fresh or frozen dissociated whole lung cells of control or IL-1β-stimulated mice using RNeasy kit (QIAGEN). Reverse transcription to cDNA was performed with the High-Capacity cDNA Reverse Transcription Kit (Thermofisher) and quantitative real-time PCR was carried out using the PowerUp SYBR Green Master Mix (Thermofisher) and Quant-Studio 7 Pro Real-Time PCR System (Thermofisher) according to the manufacturer's instructions with the primers listed below. All transcripts were quantified relative to housekeeping genes (average CT values of *Gapdh*, *18s*, and *Hprt1*) using the $2^{-\Delta CT}$ method.

Primer sequences used: (5' -> 3')
*Vcam1*:
For: TCTTGGGAGCCTCAACGGTA
Rev: CAAGTGAGGGCCATGGAGTC
*Icam1*:
For: GTGCAATCATGGTTCAGTGC
Rev: GTGTGGTGTTGTGAGCCTAT
*Gapdh*:

For: TCGTGGATCTGACGTGCCGCCTG
Rev: CACCACCCTGTTGCTGTAGCCGTA
*18s*:
For: TTGACGGAAGGGCACCACCAG
Rev: GCACCACCACCCACGGAATCG
*Hprt1*:
For: TCAGTCAACGGGGGACATAAA
Rev: GGGGCTGTACTGCTTAACCAG

## HUVEC culture and morphology analysis

Human umbilical vein endothelial cells (HUVECs) were cultured to 60–70% confluency in complete M199 media with concentrated lentiviral preparation at 37 °C for 48 h. For expression of GFP-progerin and GFP-lamin A control constructs, cells were treated with 1 µg/mL doxycycline (Sigma-Aldrich) in complete M199 media or Endothelial Cell Growth Medium 2 (EGM; Promocell) containing a low FCS serum concentration of 2%, for 4 days prior to experiments. The transduction efficiency (ratio of GFP⁺ nuclei over total number of nuclei) was evaluated at ~50% for both EGFP-lamin A and EGFP-progerin cultures.

### Confocal microscopy
For in vitro imaging, HUVECs were seeded on cover slips or on Ibidi µ-Slide V10.4 cell microscopy chambers (Ibidi, Gräfelfing, Germany) coated with porcine gelatine (0.17 mm, VWR), fixed with 4% PFA for 15 min, permeabilized with 0.2% Triton X-100 for 8 min and incubated with 3% BSA for 15 min. Cells were sequentially incubated with relevant primary and secondary antibodies and cover slips were mounted with ProLongGold (ThermoFisher Scientific) mounting medium on glass microscope slides. Confocal images were acquired using a Zeiss LSM800 laser-scanning confocal microscope as described above and analyzed by IMARIS or/and ImageJ software (NIH, USA).

### HUVEC morphology
EC body size was determined by manually tracing whole cell borders as delineated by junctional VE-Cadherin immunostaining using ImageJ and calculating the area within.

## Neutrophil adhesion and function on HUVECs

HUVECs were plated on Ibidi µ-Slide V10.4 cell microscopy chambers. Confluent cell monolayers were pre-treated with IL-1β (0.3 ng/mL for 4 h) before being co-incubated with primary human neutrophils. Human neutrophils were freshly isolated with double-density gradient of Ficoll-Histopaque (d = 1.077 and d = 1.119, Sigma) as described (Perretti et al, 1996). Briefly, 0.2 × 10⁶ neutrophils (1 × 10⁶/mL in Dulbecco's PBS (DPBS) supplemented with calcium, magnesium (PBS + +) and 1% BSA) were added to each Ibidi chamber and incubated for 5 min at 37 °C. Following incubation, the chambers were flushed with DPBS + + for 1 min at 0.1 Pa to remove non-adherent neutrophils. Chamber slides were viewed using a Nikon Eclipse TE3000 (Nikon, Tokyo, Japan), 6 images per well acquired using a Q-Imaging Retiga EXi Digital Video Camera (Q-Imaging, Surrey, Canada), and the number of adherent neutrophils was counted off-line. Subsequently, cells were fixed in 1% PFA in PBS for 1–2 h and washed with PBS prior to immunostaining and confocal microscopy.

### ROS assay
Isolated human neutrophils (100,000 cells/200 µL) were preloaded with 5 µM CellRox (Thermofisher) for 15 min and incubated on IL-1β-stimulated HUVECs for 5 min. Non-adherent cells were removed and cells were fixed, immunostained and imaged by confocal microscopy as described above. Levels of intracellular ROS mean fluorescence intensity (MFI) were quantified and the proportion of ROS-positive neutrophils manually counted using IMARIS software.

## Quantification of chemokine content in HUVECs

Control and IL-1β-stimulated HUVECs were homogenized in PBS containing 0.1% Triton and 1% Halt Protease and Phosphatase Inhibitor cocktail (Thermofisher). Cytokine and chemokine expression levels were analyzed using a Proteome Profiler Human Cytokine Array Kit (R&D Systems) as per manufacturer's instructions. The IL-8 content was further validated by ELISA (Invitrogen, sensitivity: 2 pg/mL).

## Quantification of cellular senescence of HUVECs and mouse lung endothelial cells (MLEC)

### Quantification of senescence in primary MLECs
Dissociated lung cells (as described above) were incubated in suspension in 2 mL of F12 Nutrient Mixture media (Gibco) containing 100 nM bafilomycin (ThermoFisher Scientific) for 1 h at 37 °C to increase intracellular pH. Suspended cells were then supplemented with 33 µM of $C_{12}FDG$ (ThermoFisher Scientific) for 2 h prior to analysis of $C_{12}FDG$ mean fluorescence intensity (MFI) by flow cytometry.

### Quantification of senescence in transduced HUVECs
Lentivirally transduced HUVECs described above were processed using the Senescence β-Galactosidase Staining Kit (Cell Signaling Technology, Danvers, MA, USA) following the manufacturer's instructions. Using an inverted Leica DMi8 microscope, 6–8 images per condition were acquired and analyzed offline with ImageJ.

## Immunoblotting

MLECs and HUVECs were lysed in 1x Laemmli Buffer (Sigma) supplemented with IGEPAL® CA-630 (Sigma) and 1% Halt Protease and Phosphatase Inhibitor cocktail (Thermo Fisher Scientific), then denatured at 95 °C prior to standard immunoblotting for the following proteins: mouse progerin, β-Actin, GFP, Histone H2B followed by detection with appropriate HRP-conjugated secondary antibodies (Agilent Technologies) and ECL visualized with Radiance plus Femtogram HRP substrate (Azure Biosystems) using the c600 imaging system (Azure Biosystems).

## Statistics

Data analysis was performed using Prism 9 (GraphPad) with the exception of NanoString data (see relevant section above). Results were expressed as mean ± SEM and the *n* values for each data set are provided in the figure legends. Statistical significance was assessed by two-tailed Student's t test, or one-way or two-way ANOVA followed by Tukey's post hoc test unless specified

otherwise. A *P* value less than 0.05 was considered significant. Sample size estimation was conducted through power calculation using G*power software, with input from our local statisticians. The aim was to reach a minimal level of significance of approximately $p < 0.05$ and 80% power.

## Study approval

Human blood was taken according to local research ethics committee approval (QMERC2019/83). Informed consent was provided according to the Declaration of Helsinki.

All in vivo experiments were conducted under the United Kingdom legislation according to the Animal Scientific Procedures Act 1986, with all procedures being conducted in accordance with United Kingdom Home Office regulations (P874F4263 and PP1851234).

## Data availability

NanoString gene expression data are available from the corresponding author on reasonable request.

The source data of this paper are collected in the following database record: biostudies:S-SCDT-10_1038-S44319-024-00182-x.

## Peer review information

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

## Acknowledgements

The Graphical Synopsis was created with BioRender.com. This work was principally funded by the Wellcome Trust (098291/Z/12/Z and 221699/Z/20/Z to SN). LR was supported by funding from the British Heart Foundation (FS/IBSRF/22/25121). LR and NR were additionally supported by funding from the People Programme (Marie Curie Actions) of the European Union's Seventh Framework Programme (FP7/2007–2013) under REA grant agreement no. 608765 (PCOFUND-GA-2013-608765). CGM was sponsored by Action Medical Research (GN 2272). This research was supported by the BCI Flow Cytometry Facility "CRUK FLOW CYTOMETRY CORE SERVICE GRANT at Barts Cancer Institute (Core Award C16420/A18066)" and the CMR Advanced Bio-Imaging Facility, which has been established through generous funds from the Wellcome Trust (081172/Z/06/Z, 098291/Z/12/Z, 221699/Z/20/Z) and the British Heart Foundation (IG/17/2/32993).

## Author contributions

**Loïc Rolas**: Data curation; Formal analysis; Supervision; Funding acquisition; Validation; Investigation; Visualization; Methodology; Writing—original draft; Writing—review and editing. **Monja Stein**: Data curation; Formal analysis; Validation; Investigation; Visualization; Methodology; Writing—original draft. **Anna Barkaway**: Formal analysis; Investigation; Methodology. **Natalia Reglero-Real**: Formal analysis; Investigation. **Elisabetta Sciacca**: Data curation; Formal analysis; Validation. **Mohammed Yaseen**: Investigation. **Haitao Wang**: Formal analysis. **Laura Vazquez-Martinez**: Investigation. **Matthew Golding**: Supervision; Methodology. **Isobel A Blacksell**: Resources. **Meredith J Giblin**: Data curation. **Edyta Jaworska**: Investigation. **Cleo L Bishop**: Supervision. **Mathieu-Benoit Voisin**: Supervision. **Carles Gaston-Massuet**: Resources. **Liliane Fossati-Jimack**: Formal analysis; Supervision. **Costantino Pitzalis**: Resources. **Dianne Cooper**: Resources; Supervision; Investigation. **Thomas D Nightingale**: Resources. **Carlos Lopez-Otin**: Resources. **Myles J Lewis**: Resources; Supervision; Validation. **Sussan Nourshargh**: Conceptualization; Resources; Supervision; Funding acquisition; Validation; Visualization; Methodology; Writing—original draft; Project administration; Writing—review and editing.

Source data underlying figure panels in this paper may have individual authorship assigned. Where available, figure panel/source data authorship is listed in the following database record: biostudies:S-SCDT-10_1038-S44319-024-00182-x.

## Disclosure and competing interests statement

The authors declare no competing interests.

# Expanded View Figures

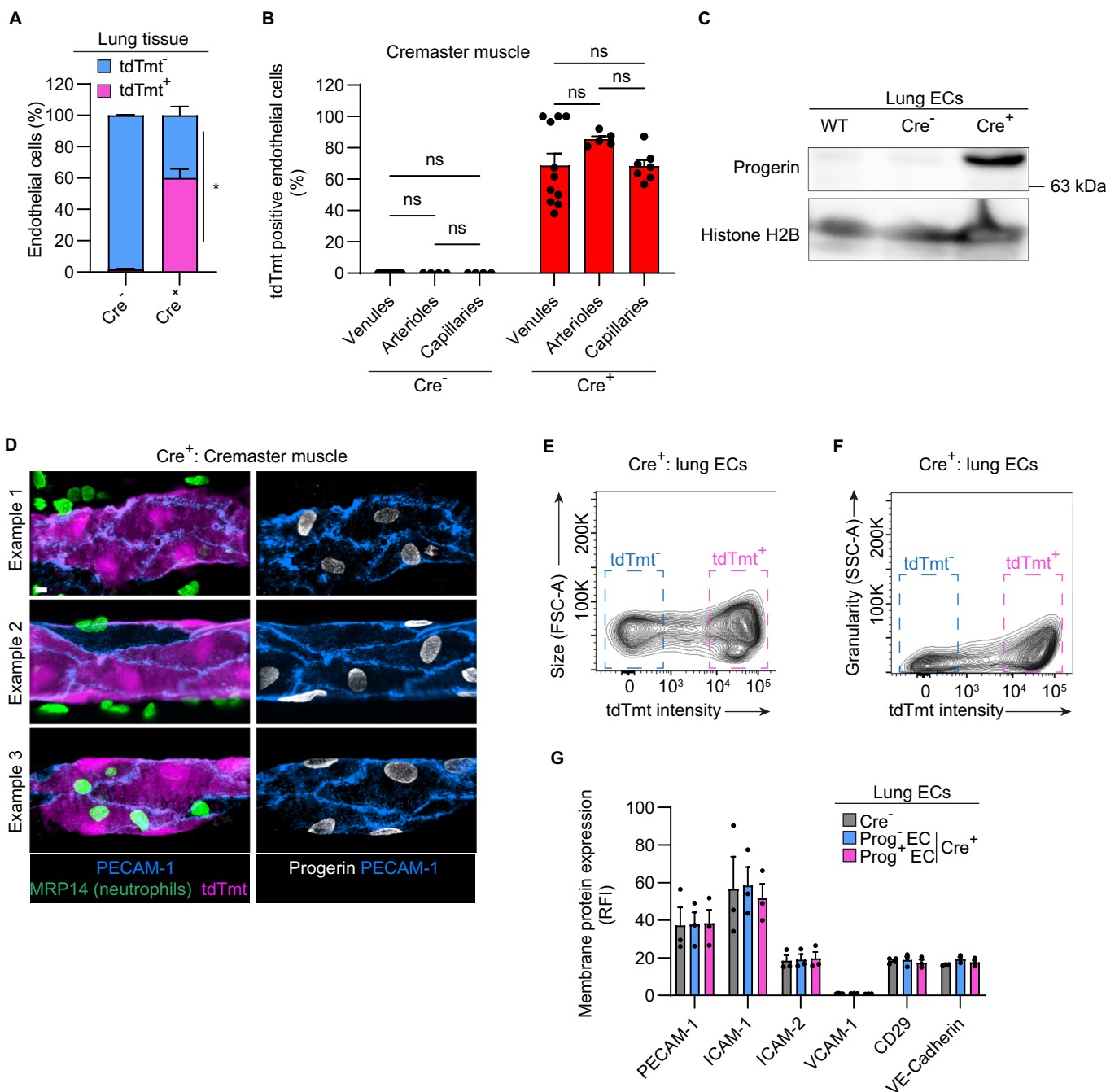

**Figure EV1. Cellular and molecular characteristics of tdTmt-progerin-expressing endothelial cells.**

(A, B) The percentage of tdTomato positive ECs in Cre⁻ and Cre⁺ mice was quantified in (A) lung ECs by flow cytometry ($n = 4$–21 mice/group in 4 independent experiments) and (B) cremaster microvasculature ECs by confocal microscopy ($n = 4$–11 mice/group in 4 independent experiments). (C) Representative immunoblot for progerin and Histone H2B (loading control) of flow cytometry sorted lung ECs from WT, Cre⁻ and Cre⁺. (D) Representative confocal images of cremaster microvasculature of acutely inflamed (IL-1β) Cre⁻ or Cre⁺ mice. Anti PECAM-1 mAb was injected i.s. and fixed tissues were immunostained for MRP14 (neutrophils) and progerin. Scale bar: 5 μm. (E–G) Lung ECs from Cre⁺ mice were analyzed by flow cytometry, and (E, F) representative plots presented according to tdTomato fluorescence intensity, depicting EC (E) size (FSC-B) and (F) granularity (SSC-A). (G) Surface expression of selected EC proteins quantified and expressed as RFI ($n = 3$–4 mice/group in 6 independent experiments). Data information: (A, B, G) Data are mean ± SEM. *$P < 0.05$; ns: not significant. (A) Two-tailed paired Student's $t$-test. (G) One-way and (B) two-way ANOVA test followed by Tukey's post hoc test.

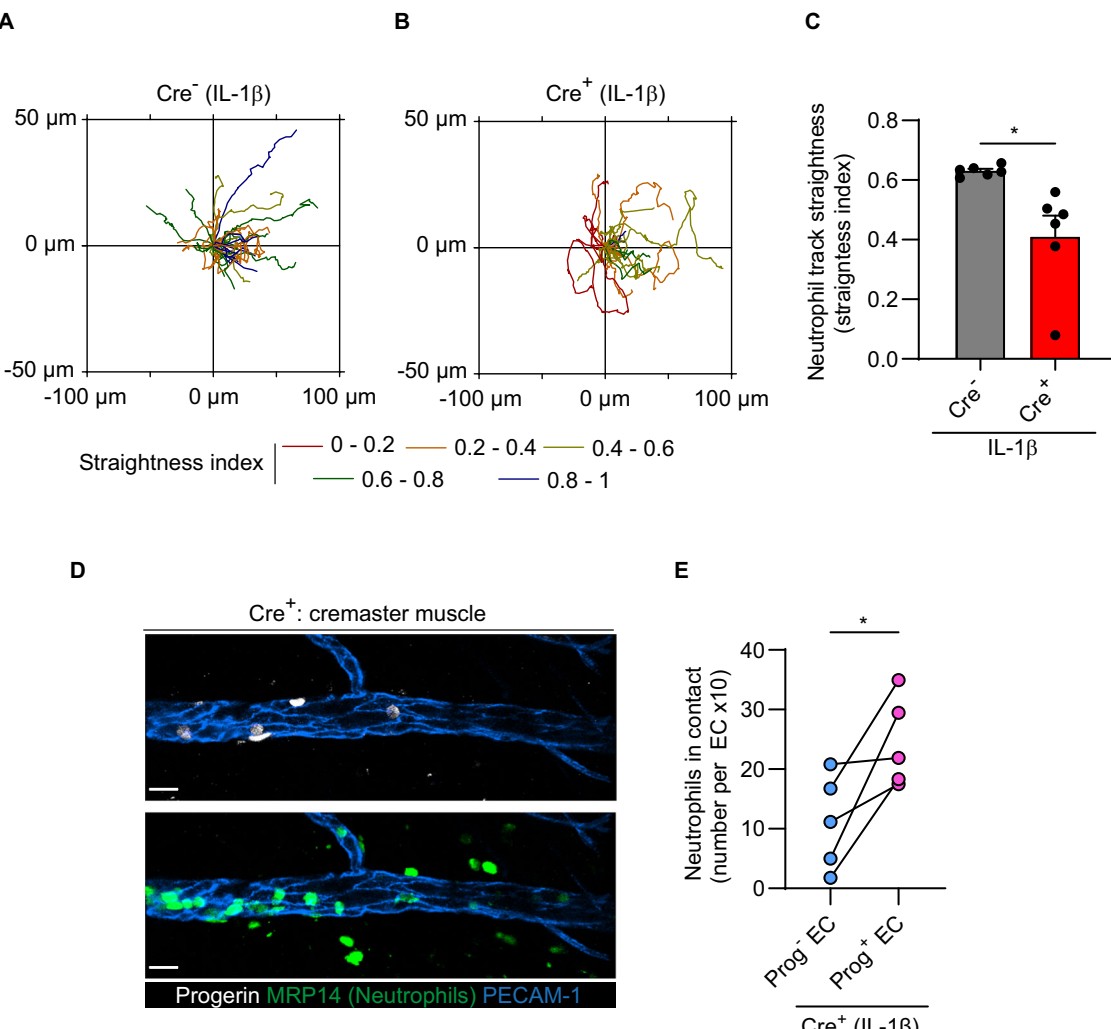

**Figure EV2. EC progerin expressing mice exhibit dysregulated luminal neutrophil crawling in IL-1β-stimulated cremasteric venules.**

Cremaster microvasculature of *Tie2-Cre;Lmna*^LCS/LCS^;*Rosa26*^tdTmt/+^;*Lyz2-EGFP-ki* Cre⁻ or Cre⁺ mice were acutely inflamed (IL-1β) and analyzed by confocal IVM. (A, B) Crawling profiles of neutrophils on ECs (responses from 28 to 34 neutrophils are displayed). (C) Luminal neutrophil track straightness (straightness index; displacement/ track length) ($n = 6$ mice/group in 12 independent experiments). (D, E) Cremaster muscles of *Tie2-Cre;Lmna*^LCS/LCS^;*Rosa26*^+/+^ Cre⁺ mice were acutely inflamed (IL-1β). (D) Fixed cremaster muscles were immunostained for progerin, MRP14 (neutrophils) and PECAM-1 (ECs), and (E) number of luminal neutrophils per progerin negative or positive EC in the same vessel segment was quantified ($n = 4$–7 mice/group in 4 independent experiments). Data information: (C) Data are mean ± SEM. *$P < 0.05$. Two-tailed unpaired Student's *t*-test. (E) *$P < 0.05$. Two-tailed paired Student's *t*-test.

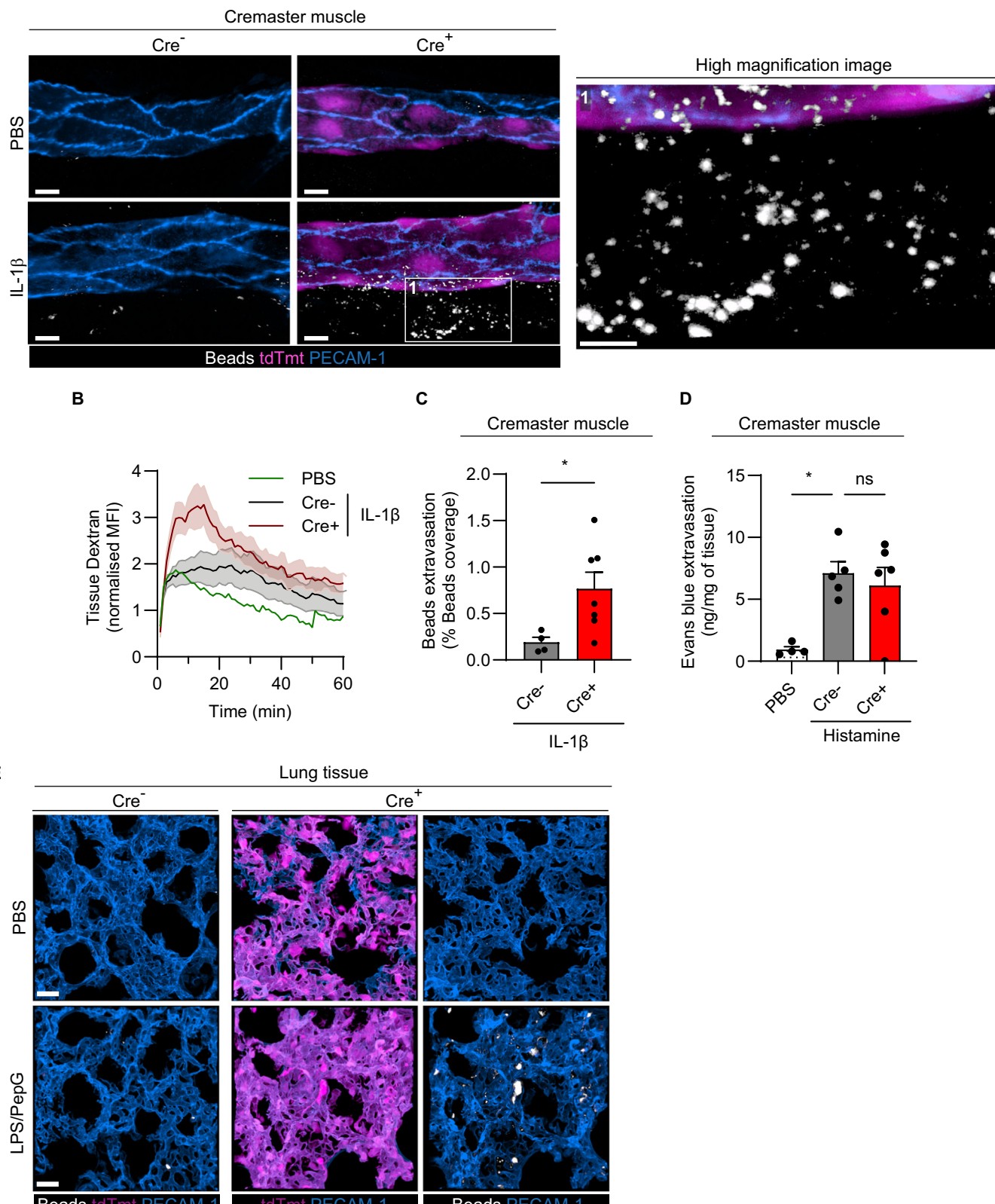

**A**

Cremaster muscle

Cre⁻ | Cre⁺

PBS

IL-1β

Beads tdTmt PECAM-1

High magnification image

**B**

Tissue Dextran (normalised MFI) vs Time (min)

PBS
Cre-  ⎤
Cre+  ⎦ IL-1β

**C**

Cremaster muscle

Beads extravasation (% Beads coverage)

Cre⁻   Cre⁺

IL-1β

**D**

Cremaster muscle

Evans blue extravasation (ng/mg of tissue)

PBS   Cre⁻   Cre⁺

Histamine

**E**

Lung tissue

Cre⁻ | Cre⁺

PBS

LPS/PepG

Beads tdTmt PECAM-1    tdTmt PECAM-1    Beads PECAM-1

◀   **Figure EV3.   EC progerin-expressing mice exhibit enhanced microvascular permeability in IL-1β-stimulated cremaster muscles.**

(A, B) Cremaster microvasculature of *Tie2-Cre;Lmna$^{LCS/LCS}$;Rosa26$^{tdTmt/+}$* Cre⁻ or Cre⁺ mice were acutely inflamed with IL-1β or PBS (control) and analyzed by confocal microscopy. Anti-PECAM-1 mAb was injected i.s. and fluorescent beads or 70-kDA dextran were injected i.v. (A) Representative confocal images of control (PBS) and stimulated (IL-1β) post-capillary venules illustrating accumulation of extravasated beads in the Cre⁺ sample. White box is magnified and displayed in the right panel. Scale bar: 5 μm. (B) Time course of dextran accumulation in the perivascular region of a selected postcapillary venule ($n = 4$–7 mice/group in 18 independent experiments). (C) Quantification of microvascular leakage calculated as percentage coverage of extravasated fluorescent beads (percentage coverage) in *Tie2-Cre;Lmna$^{LCS/LCS}$;Rosa26$^{+/+}$* Cre⁻ and Cre⁺ IL-1β-stimulated tissues ($n = 4$–7 mice/group in 4 independent experiments). (D) Vascular leakage in the cremaster muscle of *Tie2-Cre;Lmna$^{LCS/LCS}$;Rosa26$^{tdTmt/+}$* Cre⁻ or Cre⁺ mice injected i.s. with PBS, or histamine (30 min) ($n = 4$–7 mice/group in 7 independent experiments). (E) Lungs from Cre⁻ and Cre⁺ mice treated i.p. with PBS (control) or LPS/PepG were analyzed by confocal microscopy for vascular leakage (extravasation of i.v. injected fluorescent microbeads). Representative confocal images of lung tissue sections immunostained for PECAM-1 (ECs) illustrate increased bead extravasation (white) in close apposition to tdTmt positive vessel segments of Cre⁺ mice. Scale bar: 30 μm. Data information: (C, D) Data are mean ± SEM. *$P < 0.05$; ns: not significant. (C) Two-tailed unpaired Student's *t*-test and (D) one-way ANOVA followed by Tukey's post hoc test.

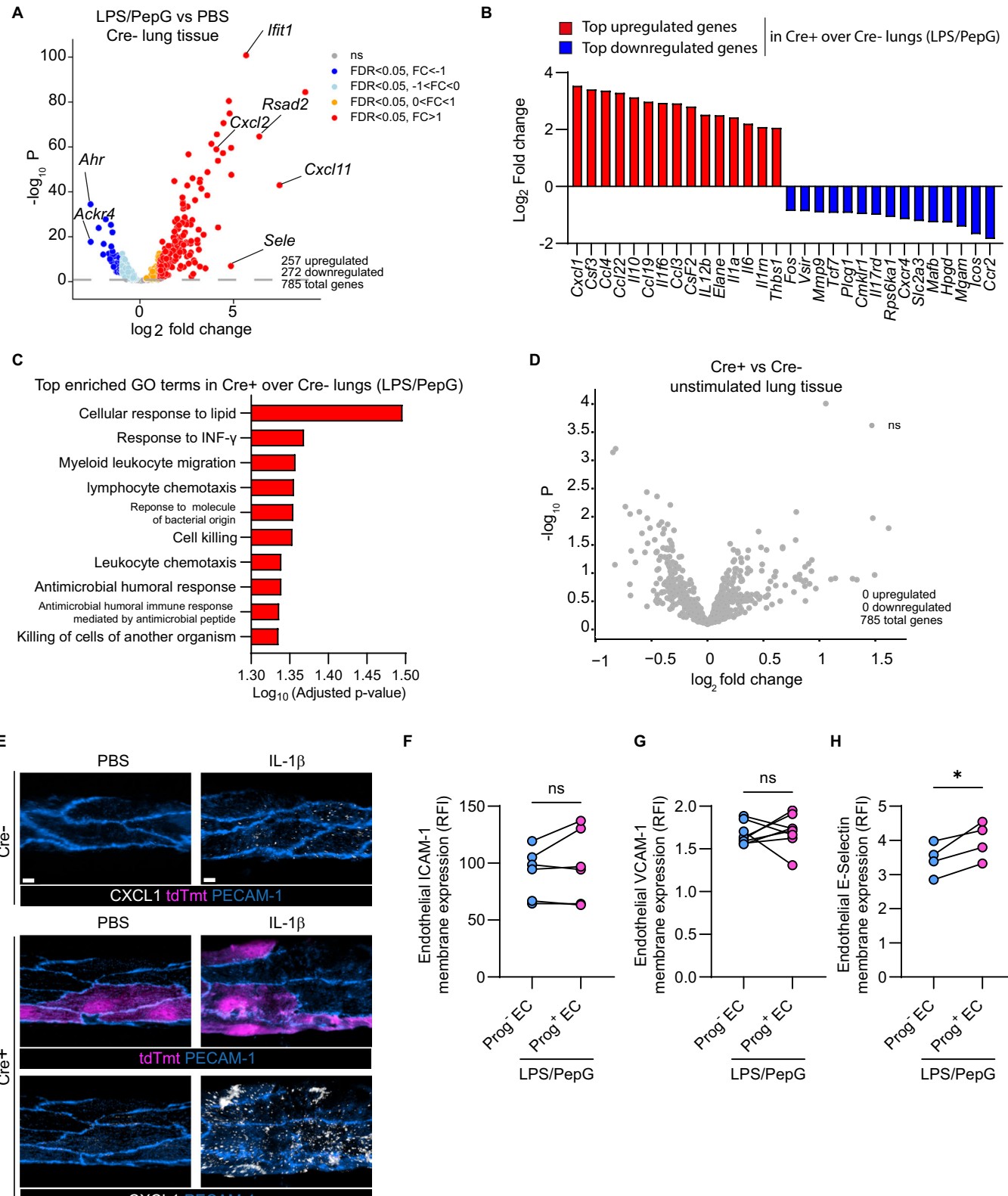

**Figure EV4. Molecular characteristics of inflamed tissues of EC progerin expressing mice.**

(A–C) Cre⁻ and Cre⁺ mice were stimulated with i.p. PBS or LPS/PepG (2 h) after which tissue samples were analyzed by targeted NanoString transcriptomics. (A) Volcano plot of the relative difference in gene expression in whole lung tissue of control or stimulated Cre⁻ mice as assayed by NanoString (785 genes; $n = 6$ mice/group in 1 experiment). Significantly differentially expressed genes (FDR > 0.05) are color coded as follows: Downregulated genes in dark blue (fold change, FC < −1) and light blue (−1 < FC < 0); upregulated genes in orange (0 < FC < 1) and red (FC > 1). (B, C) Gene expression profiles in inflamed lung tissues of Cre⁺ mice, relative to Cre⁻. (B) The 15 most highly upregulated (red) and highly downregulated (blue) genes in Cre⁺ mice are displayed (785 genes; $n = 3$ mice/group in 1 experiment). (C) Gene Ontology (GO) terms enrichment analysis of 785 differentially expressed genes (DEGs) in inflamed lungs of Cre⁺ mice (scored as −$\log_{10}$ (p-value)). The presented results are based on Fisher's exact test with false discovery rate adjustment. (D) Volcano plot of the relative difference in gene expression in whole lung tissue of unstimulated Cre+ vs Cre-mice as assayed by NanoString (785 genes; $n = 3$ mice/group in 1 experiment). (E) Representative confocal images of cremaster muscles treated with PBS (control) or IL-1β-stimulated cremaster muscles of Cre⁻ and Cre⁺ mice. Tissues were immunostained for PECAM-1 and CXCL1, with tdTomato fluorescence identifying progerin expressing ECs. Scale bar: 5 μm. (F–H) Surface expression of (F) ICAM-1, (G) VCAM-1, and (H) E-Selectin, presented as RFI, on progerin negative and positive lung ECs derived from Cre⁺ mice subjected to LPS/PepG stimulation ($n = 4$–6 mice/group in 6 independent experiments). Data information: (F–H) *$P < 0.05$; ns: not significant. Two-tailed paired Student's $t$-test.

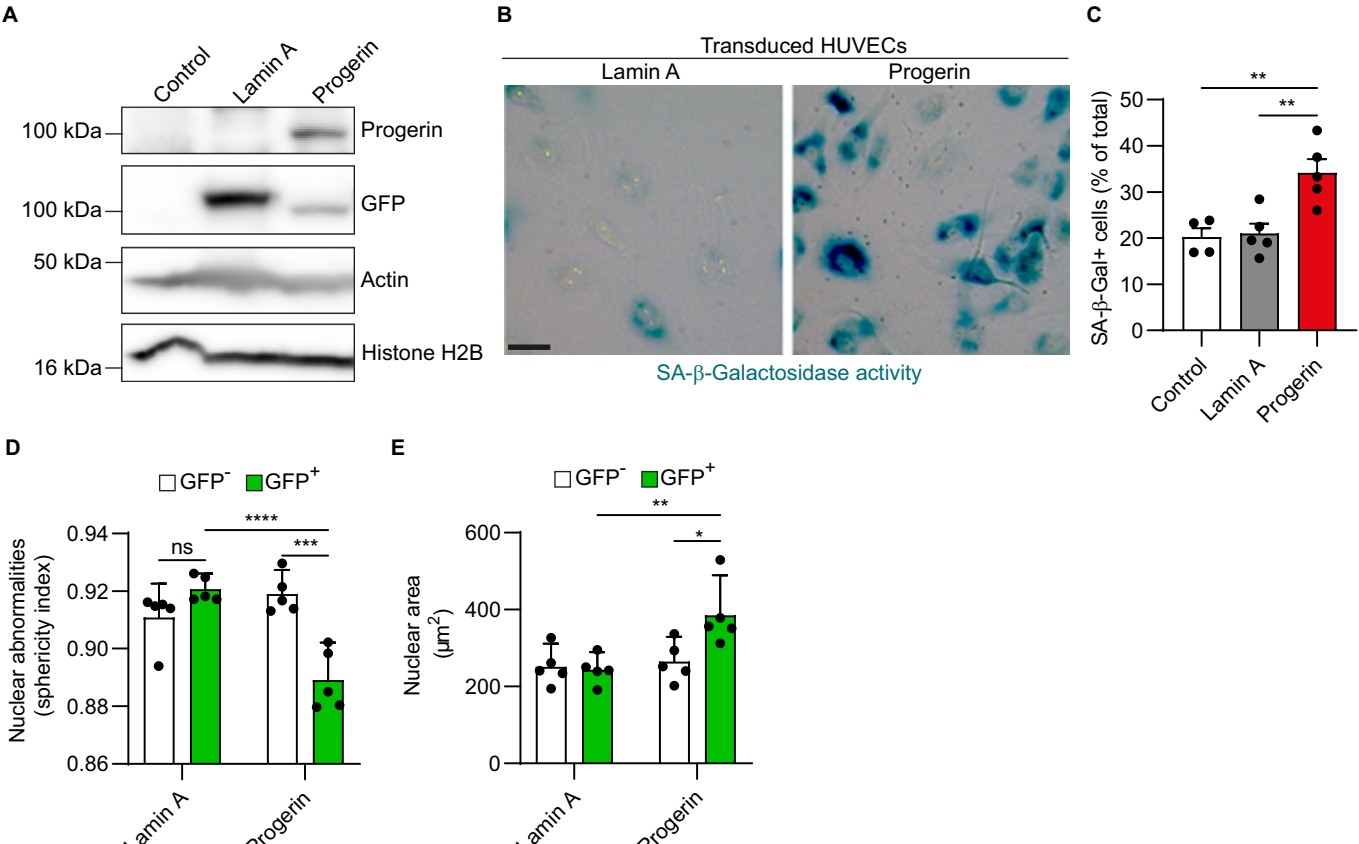

**Figure EV5. Progerin expressing HUVECs exhibit cellular senescence and a pro-secretory phenotype.**

HUVECs were transduced with GFP-tagged lamin A or GFP-tagged progerin expressing lentiviral constructs. (A) Representative immunoblot of control and transduced HUVECs probed for progerin, GFP, β-Actin, and Histone-H2B proteins. (B) Representative bright field image of transduced HUVECs assayed for SA-β-Gal activity (blue) and (C) its associated quantification ($n = 4$–5 in 5 independent experiments). Scale bar: 40 μm. (D, E) Non-transduced (GFP-) and transduced (GFP+) HUVECs ($n = 5$ in 5 independent experiments) were quantified for (D) nuclear shape abnormalities and (E) nuclear size. Data information: (C–E) Data are mean ± SEM. *$P < 0.05$; **$P < 0.01$; ***$P < 0.001$; ****$P < 0.0001$; ns: not significant. (C) One-way and (D, E) two-way ANOVA followed by Tukey's post hoc test.

