## [Peer Review File · EMBO Reports]

Senescent endothelial cells promote pathogenic neutrophil trafficking in inflamed tissues

Loïc Rolas, Monja Stein, Anna Barkaway, natalia Reglero-Real, Elisabetta Sciacca, Mohammed Yaseen, Haitao Wang, Laura Vazquez-Martinez, Matthew Golding, Isobel Blacksell, Meredith Giblin, Edyta Jaworska, Cleo Bishop, Mathieu Voisin, Carles Gaston-Massuet, Liliane Fossati-Jimack, Costantino Pitzalis, Dianne Cooper, Thomas Nightingale, Carlos Lopez-Otin, Myles Lewis, and Sussan Nourshargh

Corresponding author(s): Sussan Nourshargh (s.nourshargh@qmul.ac.uk)

Review Timeline:

Submission Date:	31st Oct 23
Editorial Decision:	7th Dec 23
Revision Received:	19th Apr 24
Editorial Decision:	16th May 24
Revision Received:	28th May 24
Accepted:	7th Jun 24

Editor: Deniz Senyilmaz Tiebe

Transaction Report:

Dear Prof. Nourshargh,

Thank you for the submission of your research manuscript to our journal, which was now seen by three referees, whose reports are copied below.

My apologies for this unusual delay in getting back to you. It took longer than anticipated to receive the full set of referee reports.

Referees express interest in the proposed role of endothelial cell senescence in regulation of neutrophil infiltration. However, they also raise concerns that need to be addressed to consider publication here.

Given these positive recommendations, we would like to invite you to submit a revised manuscript. Please revise your manuscript with the understanding that the referee concerns (as in their reports) must be fully addressed and their suggestions taken on board. Please address all referee concerns in a complete point-by-point response. Acceptance of the manuscript will depend on a positive outcome of a second round of review. It is EMBO reports policy to allow a single round of major experimental revision only and acceptance or rejection of the manuscript will therefore depend on the completeness of your responses included in the next, final version of the manuscript.

We realize that it is difficult to revise to a specific deadline. In the interest of protecting the conceptual advance provided by the work, we recommend a revision within 3 months. Please discuss the revision progress ahead of this time with me if you require more time to complete the revisions, or if you have questions or comments regarding the revision (also by video chat).

1. A data availability section providing access to data deposited in public databases is missing (where applicable).
2. Your manuscript contains statistics and error bars based on $n=2$. Please use scatter plots in these cases.

You can submit the revision either as a Scientific Report or as a Research Article. For Scientific Reports, the revised manuscript can contain up to 5 main figures and 5 Expanded View figures, and it should not exceed 27000 characters. If the revision leads to a manuscript with more than 5 main figures it will be published as a Research Article. In this case the Results and Discussion section should be separate. If a Scientific Report is submitted, these sections have to be combined. This will help to shorten the manuscript text by eliminating some redundancy that is inevitable when discussing the same experiments twice. In either case, all materials and methods should be included in the main manuscript file.

4) a .docx formatted letter INCLUDING the reviewers' reports and your detailed point-by-point responses to their comments. As part of the EMBO publication's Transparent Editorial Process, EMBO reports publishes online a Review Process File (RPF) to accompany accepted manuscripts. This File will be published in conjunction with your paper and will include the referee reports, your point-by-point response and all pertinent correspondence relating to the manuscript.

<https://www.embopress.org/page/journal/14693178/authorguide#transparentprocess>

5) a complete author checklist, which you can download from our author guidelines <https://www.embopress.org/page/journal/14693178/authorguide>. Please insert information in the checklist that is also reflected in the manuscript. The completed author checklist will also be part of the RPF.

6) Please note that all corresponding authors are required to supply an ORCID ID for their name upon submission of a revised manuscript (<<https://orcid.org/>>). Please find instructions on how to link your ORCID ID to your account in our manuscript tracking system in our Author guidelines <<https://www.embopress.org/page/journal/14693178/authorguide#authorshipguidelines>>

7) Before submitting your revision, primary datasets produced in this study need to be deposited in an appropriate public database (see <https://www.embopress.org/page/journal/14693178/authorguide#datadeposition>). Please remember to provide a reviewer password if the datasets are not yet public. The accession numbers and database should be listed in a formal "Data Availability" section placed after Materials & Method (see also <https://www.embopress.org/page/journal/14693178/authorguide#datadeposition>). Please note that the Data Availability Section is restricted to new primary data that are part of this study. * Note - All links should resolve to a page where the data can be accessed. *
If your study has not produced novel datasets, please mention this fact in the Data Availability Section.

Additional information on source data and instruction on how to label the files are available:
<https://www.embopress.org/page/journal/14693178/authorguide#sourcedata>

9) Our journal encourages inclusion of *data citations in the reference list* to directly cite datasets that were re-used and obtained from public databases. Data citations in the article text are distinct from normal bibliographical citations and should directly link to the database records from which the data can be accessed. In the main text, data citations are formatted as follows: "Data ref: Smith et al, 2001" or "Data ref: NCBI Sequence Read Archive PRJNA342805, 2017". In the Reference list, data citations must be labeled with "[DATASET]". A data reference must provide the database name, accession number/identifiers and a resolvable link to the landing page from which the data can be accessed at the end of the reference. Further instructions are available at <http://www.embopress.org/page/journal/14693178/authorguide#referencesformat>

- the name of the statistical test used to generate error bars and P values,
- the number (n) of independent experiments (please specify technical or biological replicates) underlying each data point,
- the nature of the bars and error bars (s.d., s.e.m.),
- If the data are obtained from n Program fragment delivered error `Can't locate object method "less" via package "than" (perhaps you forgot to load "than"?) at //ejpvfs23/sites23b/embor_www/letters/embor_decision_revise_and_review.txt line 56.' 2, use scatter blots showing the individual data points.

12) Please also note our reference format:

I look forward to seeing a revised version of your manuscript when it is ready. Please let me know if you have questions or comments regarding the revision.

Kind regards,

Deniz Senyilmaz Tiebe

Deniz Senyilmaz Tiebe, PhD
Scientific Editor
EMBO Reports

Referee #1:

The study by Rolas et al., explores a potential correlation between endothelial senescence and inflammation. For this purpose, they employ a new endothelial-specific progeroid mouse model, where progerin-expressing ECs express hallmarks of senescence such as the presence of phosphorylated histone H2AX, SA- β -Galactosidase activity and increased cell size and granularity, accompanied by increased expression of ICAM and VCAM. Using a tdTomato-reporter, and a newly developed progerin antibody, neutrophil adhesion to individual progerin-expressing ECs could be followed by intravital imaging. The authors show that IL1 β -stimulated adhesion and crawling of neutrophils was considerably prolonged to progerin-expressing ECs in an CXCL1-dependent manner. Similar results were obtained using a mouse model for LPS/PepG-induced lung damage and progerin-expressing human ECs. Transcriptome analyses showed a proinflammatory expression profile specifically in progerin+ ECs in the damaged lungs. This is a high-quality, impressive presentation with novel, important findings.

I have only one comment which the authors could bring up in some more detail, in the discussion: The authors show that mice depleted of their circulating neutrophils display a complete inhibition of microbead extravasation across inflamed Cre+ cremasteric venules (Fig. 4C). It was not quite clear to this reviewer whether the authors refer to leakage in conjunction with neutrophil extravasation or leakage at separate sites? Is neutrophil-independent vascular permeability also elevated in the progerin-expressing venules? This is a relevant question as the connection between inflammation and vascular permeability induced by directly acting agonists such as proinflammatory cytokines remains to be resolved. Furthermore, directly induced vascular leakage is believed to occur at sites distinct from where inflammatory cells extravasate. The authors touch upon this question in the Discussion, 450-463, but this text is relatively vague and could address the issues more directly.

Referee #2:

This is an exceptional manuscript documenting the *in vivo* effects of senescent endothelial cells to regulate neutrophil crawling and adhesion.

The report shows real time imaging with high level quantification and mechanistic studies to define at least one chemokine responsible for the effect.

The authors should show whether CXCL1 is specific for progeria or is a general mechanism. Is CXCL1 also elevated in senescent endothelial cells induced by other stimulants, for example *in vitro* on H₂O₂ stimulated ECs, and whether this also regulates neutrophil attachment.

The authors should comment on the increased ICAM and VCAM mRNA in the lung ECs (Figure 2 I,J) yet no enhanced protein expression (EV2D). Are they primed and hence respond to lower levels of inflammatory stimulation? This would be important to know for understanding the role of senescent ECs.

EV1-How were the venules, arterioles and capillaries differentiated?

Figure 4- Permeability is also measured by biotin or albumin leakage (eg in Ting K et al 2023 Geroscience). What is the size of the beads and is the difference in organ bed or a leak size difference, since they show basal permeability changes at sites of senescence.

I believe the supplementary Figures are out of order to what is listed in the text.

Referee #3:

The authors use an elegant genetic model in the mouse with EC-specific expression of the senescence associated protein

progerin. They show that the expression of progerin in EC leads to enhanced intravascular neutrophil adhesive properties and increased (neutrophil dependent) vascular permeability in IL-1b stimulated cremaster muscle venules and LPS/PG stimulated lungs. They link these effects particularly on a strong progerin dependent induction of EC-specific expression of CXCL1. Using lentiviral transduced HUVECs for expression of progerin, they are able to recapitulate their findings from the mouse in primary human endothelial cells. Overall, a well done study, which gives us new insights of how EC-expressed progerin modulates immune responses.

I have several major and minor points, which might be worth to be addressed by the authors:

Major:

- Fig.3) Please indicate how EC-specific progerin expression in IL-1b stimulated cremaster muscle venules affects neutrophil extravasation. Does expression of tdTmt alone influence neutrophil adhesive behavior?
- Fig.4) Again it would be nice to show that tdTmt alone has no influence on vascular permeability.
- Fig. 7 and 8) IL-1b also upregulates E-selectin in cremaster muscle venules. Previous work (Smith et al. JEM 2004) has demonstrated that chemokines and E-selectin function in a redundant manner in mediating neutrophil recruitment. In your data, blocking CXCL1 has a profound effect on neutrophil adhesive function in the EC-progerin group. How do you explain these findings? What do you see in unstimulated cremaster muscle venules by intravital microscopy? As shown in your cytokine profiles (Fig. 8) for HUVEC, CXCL8 is very high already under baseline conditions. This might be the same in the mouse for CXCL1 and also impact neutrophil recruitment in unstimulated cremaster muscles in the presence of progerin.
- Formally, the authors have shown that EC-specific progerin influences neutrophil recruitment during the inflammatory response. How important is progerin for senescence processes in aged mice and men? Did you try to stain for EC-progerin in aged mice (stimulated and unstimulated)? An upregulation of progerin together with CXCL1 in EC of aged mice would provide an important link btw progerin, ageing and enhanced neutrophil responses (as reported).

Minor:

- Fig.3E and C and Fig.7A, please add units (y-axis).
- Fig.6A: you might change order Cre- vs Cre+ to Cre+ vs Cre- as this is shown in the blot.
- A connection btw CXCR2 and neutrophil responses have already been reported previously. This should be cited: Nomellini V, Faunce DE, Gomez CR, Kovacs EJ. An age-associated increase in pulmonary inflammation after burn injury is abrogated by CXCR2 inhibition. *J Leukoc Biol.* 2008;83:1493-1501.
- Nomellini V, Brubaker AL, Mahbub S, Palmer JL, Gomez CR, Kovacs EJ. Dysregulation of neutrophil CXCR2 and pulmonary endothelial Icam-1 promotes age-related pulmonary inflammation. *Aging Dis.* 2012;3:234-247.

Point-by-point Response to Referee's Comments

Referee #1:

We thank the referee for their positive comments, stating that our manuscript presents high quality, novel and important findings. The referee's minor comments are addressed below:

Neutrophil-dependent versus neutrophil-independent vascular leakage: We thank the referee for raising this important point. To categorically address, the revised MS includes new data showing that mice expressing endothelial cell (EC) progerin exhibit a similar level of histamine-induced vascular permeability to that detected in control mice. Quantified by extravascular accumulation of i.v. Evans Blue, this was seen in two distinct models, namely, inflamed mouse dorsal skin and cremaster muscle (**new Figures 4D and EV3D**). These new results illustrate that senescent ECs do not exhibit enhanced neutrophil-independent vascular leakage as elicited by local histamine. Importantly, in the cutaneous model, neutrophil-dependent vascular leakage (as induced by intradermal chemoattractant LTB₄) was yet again increased (**new Figure 4D**), in line with our previous data from IL-1 β -inflamed cremaster muscles (**Figures 4A-C**). For technical reasons, the "neutrophil-dependent" responses quantified were not directly aligned with sites of neutrophil adhesion. Briefly, unlike neutrophil attachment, it is very difficult to ascertain if vascular leakage through EC junctions is mediated by Prog⁺ or Prog⁻ ECs, especially when cells of such phenotype were often directly found side-by-side. The best strategy we have employed for addressing this point in is through the use of mice with different degrees of recombination and hence % of Prog⁺ ECs, as shown in Figure 4B and also through gross association as shown in Figure 5E.

Referee #2:

We thank the referee for their supportive comments. The specific comments are addressed below:

- **Role of CXCL1:** To address the potential involvement of endothelial cell (EC) CXCL1 in other models, we chose to investigate the most relevant setting, that of physiological ageing. As such, the revised MS provides *in vivo* data from aged mice showing that ECs with a senescent phenotype (p21^{high}) express higher levels of CXCL1 and support greater level of neutrophil attachment to p21^{low} ECs (**new Figures 7E-G**). Furthermore, pharmacological treatment of aged mice with a blocking anti-CXCL1 mAb suppressed the observed excessive neutrophil adhesion, reducing it to that seen in young mice (**new Figure 7H**).
- **ICAM and VCAM mRNA:** The referee raises an interesting point. We agree that the data suggest that senescent ECs maybe primed and hence respond faster and potentially better to lower levels of inflammatory stimuli. Indeed, this is what we observed in our pathological lung injury model, i.e. faster and more excessive inflammation in EC progerin-expressing mice subjected to LPS/PepG (**Figure 5A**). However, having now quantified protein levels of ICAM-1 and VCAM-1 on stimulated lung ECs, the results show no difference between progerin-expressing and control ECs (**new Figures EV4E-F**). Interestingly, we observed a significant elevation of E-selectin protein on progerin-expressing ECs (**new Figure EV4G**), suggesting that this adhesion molecule may have a functional role. Overall, we would speculate that a potentially primed state of senescent ECs may contribute to the exaggerated inflammatory response of physiological aging. However, this phenomenon may be more significant in chronic inflammatory settings, as opposed to the acute models employed in our study. **This hypothesis is now included in the revised Discussion (p.16).**
- **Distinguishing venules, arterioles and capillaries:** This analysis has been further clarified in the **Methods (p.22)**. Briefly, the *in vivo* labelling of ECs with CD31 provides an excellent means of observing the microcirculation network and hence distinguishing arteriolar vs venular ECs by the morphology of the vessels (e.g. arterioles have thicker walls), direction of flow, EC shape (e.g. elongated in arterioles) and separating these from capillaries by their diameters. Additionally, as neutrophil-vessel wall interactions exclusively occur in stimulated post-capillary venules, this functional read-out provides a further robust means of distinguishing venules from capillaries and arterioles.
- **Quantification of vascular leakage:** Vascular leakage can indeed be quantified by multiple methods, such as local accumulation of i.v. administered beads or large proteins. In our study, we

have commonly employed fluorescent 20nm beads, using a protocol adapted from the group of Prof Donald McDonald (*Le et al., Am J Pathol., 2015*). This method is widely used by many permeability researchers and indeed different vascular beds will exhibit differential passage of tracers from blood to tissues due to the diversity of endothelial cell junctional structures in different organs (please see *Claesson-Welsh L et al., Trends Mol Med., 2021*). Of note, our findings showing enhanced neutrophil-dependent vascular leakage in EC progerin-expressing vessels using i.v. fluorescent beads are now further corroborated using two additional methods, namely using i.v. 70-kDA TRITC Dextran (new Figure EV3B) and Evans Blue (new Figure 4D).

- **Supplementary Figure numbering:** We apologize for this error; rectified in the revised MS.

Referee #3:

We are grateful for the positive comments and insightful suggestions. The referee's comments are addressed below:

Major:

- **Neutrophil extravasation:** This is indeed important, and we thank the referee for requesting the data. The revised MS now includes neutrophil extravasation results from IL-1 β -stimulated endothelial cell (EC) progerin-expressing mice, showing no difference from responses acquired in tissues from littermate control Cre⁻ mice (new Figure 3B).
- **Effect of tdTmt on neutrophil behaviour and vascular permeability:** To address this valid point we have now quantified neutrophil attachment and vascular permeability in EC-progerin expressing mice (*Tie2-Cre;Lmna^{LCS/LCS}*) that were not inter-crossed with the *Rosa tdTmt* reporter line. The results yet again show enhanced inflammatory events in EC-progerin expressing mice (new Figures EV2D-E and EV3C), corroborating our previous findings.
- **E-Selectin vs CXCL1:** Quantification of membrane E-selectin on stimulated progerin-expressing lung ECs shows evidence of enhanced protein levels (new Figure EV4G), suggesting a potential functional role, in agreement with the findings of *Smith et al., 2004*. Regarding CXCL1, it is important to note that our *in vivo* studies deliberately employed a sub-optimal dose of the anti-CXCL1 mAb, a strategy that had no discernible effect on neutrophil attachment to stimulated cremaster muscles of non-progerin expressing mice/tissues (see Figure 7A and 7B). However, this protocol suppressed the excessive adhesion of neutrophils to progerin-expressing ECs. Hence, whilst neutrophil attachment to non-senescent ECs maybe mediated by CXCL1 and other factors (e.g. E-selectin), in senescent ECs, the excessive production of CXCL1 appears to facilitate the noted enhanced adhesion.
- **HUVEC and CXCL8:** Indeed, our HUVEC data indicate moderate levels of CXCL8 in unstimulated cells, results that we believe represent activation of the cells by the *in vitro* culture conditions. Crucially, *in vivo*, unstimulated EC progerin expressing tissues exhibit low levels of CXCL1, similar to that seen in unstimulated or stimulated non-progerin expressing tissues (see Figure 6B-C). Furthermore, control (PBS-treated) EC progerin-expressing tissues do not support neutrophil attachment (data in text, p.11).
- **Progerin and senescence:** We thank the reviewer for this significant comment. Fundamentally we acknowledge that in our study we are using progerin-expressing ECs as a model of EC senescence. The reason for this is that to date, surprisingly, despite the existence of senescence-ablator transgene mouse models and senescent reporter mice, to our knowledge there is no cell-specific model of cellular senescence induction (e.g. p16/p21 over-expression). As such, the model that we have employed, based on the accelerated ageing syndrome of HGPS as driven by progerin, provides a valuable tool. Whilst we fully recognise the limitations of this approach, it is critical to note that HGPS and physiological aging share numerous common cellular and molecular features, and in this context, progerin-induced senescence is a key hallmark. These points have been covered in the Introduction and Discussion. Additionally, and in direct response to the referee's comment, the dominant negative mutation that gives rise to progerin is found at low levels in young individuals,

and at a higher incidence in aged humans in multiple tissues (e.g. skin, heart, liver, kidney, stomach and skeletal muscles, e.g. *Scaffidi et al., Science 2006*). Hence, whilst our efforts to detect progerin in inflamed tissues of aged mice have failed (likely due to low protein levels), there is strong evidence in the literature to support the hypothesis that aberrant forms of lamin A contribute to physiological ageing. This is very likely through induction of nuclear deformability and hence induction of senescence.

Despite the lack of evidence for progerin expression in aged mice, we fully agree with the referee that our overall conclusions require validation in aged mice. As such, the revised manuscript contains new data showing that in inflamed aged tissues, ECs with a senescent phenotype (i.e. p21^{high}), express higher levels of CXCL1 and support greater neutrophil attachment, as compared to p21^{low} ECs (new Figures 7E-G). Furthermore, pharmacological treatment of aged mice with a blocking anti-CXCL1 mAb suppressed the observed excessive neutrophil adhesion to the level seen in young mice (new Figure 7H).

Minor:

- **Amendments to Figures:** All changed as advised.
- **Additional References:** Cited as requested.

Dear Sussan,

Thank you for submitting your revised manuscript. It has now been seen by all of the original referees.

As you can see, the referees find that the study is significantly improved during revision and recommend publication. However, I need you to address the points below before I can accept the manuscript.

- We note a discrepancy in the spelling of one of the author names - i.e. the author's name was spelled as Dianne Cooper in the manuscript file whereas it was spelled as Diane Cooper in the manuscript submission system.
- Please rename the Conflicts of Interests section as "Disclosure Statement and Competing Interests".
- We note the phrase 'data not shown' on pages 10 and 18, which is not allowed as per journal policy. Please either show the data or remove the statements.
- Funding information needs to be a part of the Acknowledgements section. Moreover, Barts Charity and QMUL should also be entered in the manuscript submission system.
- The movie nomenclature needs to be corrected to Movie EV1, etc. and their callouts in the manuscript text also need to be corrected. Moreover, the legends need to be removed from the manuscript text and each should be provided as a readme txt file with corrected title to Movie EV1, etc.; then each movie should be zipped up together with its legend and uploaded as one folder per movie.
- During our routine checks on the source data, we noted some potential duplications in the numerical data, which needs clarification (please see attached excel files, where the potential duplications are highlighted).
- Please resubmit source data as one zip file per figure.
- Our production/data editors have asked you to clarify several points in the figure legends:
 - o Please note that a separate 'Data Information' section is required in the legends of figures 1h; 2b, d-i; 3b, d-g; 4a-d; 5a-b, d-e; 6b-c, e; 7a-d, f-h; 8b, d, f-g; EV 1a-b, g; EV 2c, e; EV 3c-d; EV 4e-g; EV 5c-e. Please see <https://www.embopress.org/page/journal/14693178/authorguide#figureformat> for an example.
 - o Please note that the legend for figure 3b is incorrectly labelled as 3a in the legend. This needs to be rectified.
 - o Please note that the legend for figures 3d-f is incorrectly labelled as 3b-d. This needs to be rectified.
 - o Please indicate the statistical test used for data analysis in the legends of figure 6a; EV 4a, c.
 - o Please note that in figure 3b; there is a mismatch between the annotated p values in the figure legend and the annotated p values in the figure file that should be corrected.
 - o Please note that information related to n is missing in the legends of figures 3b; EV 4e-g.
 - o Although 'n' is provided, please describe the nature of entity for 'n' in the legend of figure 8d.
 - o Please note that the error bars are not defined in the legends of figures 3b; 7h; 8d.
 - o Please note that the scale bar needs to be defined for figure 1b.
- The manuscript sections should be in the following order: Title page - Abstract & Keywords - Introduction - Results - Discussion - Methods - Data Availability - Acknowledgments - Disclosure Statement & Competing Interests - References - Figure Legends - Tables with legends - Expanded View Figure Legends.
- The synopsis image needs to be 550px wide and 300-600px high. When your synopsis image is resized accordingly, the labels are too small to read (please see attached). Please provide a synopsis image with larger labels.

Thank you again for giving us to consider your manuscript for EMBO Reports, I look forward to your minor revision.

Kind regards,

Deniz

--

Deniz Senyilmaz Tiebe, PhD
Editor
EMBO Reports

Referee #1:

The authors have increased the quality of their paper further, it is suitable for publication. I congratulate the authors to an excellent study.

Referee #2:

The authors have addressed all the issues raised in my review.

Referee #3:

The authors have adequately addressed all concerns raised by this reviewer including new experiments. Therefore, I advocate acceptance of the manuscript.

Point-by-Point Response to the Editor's Comments:

- ***We note a discrepancy in the spelling of one of the author names - i.e. the author's name was spelled as Dianne Cooper in the manuscript file whereas it was spelled as Diane Cooper in the manuscript submission system.***

Thank you and apologies, the correct spelling is **Dianne Cooper**. Please note that despite our attempts to amend this on the on-line submission system, the name reverts to the wrong spelling. Hope the Editorial office can help with this matter.

- ***Please rename the Conflicts of Interests section as "Disclosure Statement and Competing Interests".***

Done.

- ***We note the phrase 'data not shown' on pages 10 and 18, which is not allowed as per journal policy. Please either show the data or remove the statements.***

Thank you. We have inserted the control data aligned with the phrase on page 10 as a new Figure Panel EV4D. The phrase on page 18 in Methods has been removed.

- ***Funding information needs to be a part of the Acknowledgements section. Moreover, Barts Charity and QMUL should also be entered in the manuscript submission system.***

Barts Charity and QMUL have been removed from the Acknowledgements as the funding from these sources was deemed not to be significant for the present study.

- ***The movie nomenclature needs to be corrected to Movie EV1, etc. and their callouts in the manuscript text also need to be corrected. Moreover, the legends need to be removed from the manuscript text and each should be provided as a readme txt file with corrected title to Movie EV1, etc.; then each movie should be zipped up together with its legend and uploaded as one folder per movie.***

Done.

- ***During our routine checks on the source data, we noted some potential duplications in the numerical data, which needs clarification (please see attached excel files, where the potential duplications are highlighted).***

We would like to thank the EMBO Reports team for bringing these important errors to our attention. After conducting a detailed internal investigation, we discovered that certain quantifications that were conducted early in the project by staff who have left the team were inadvertently duplicated during copy and pasting from old files into new files. We have now meticulously organised the quantified data, removed the duplication, and restored the original data to its intended state. Importantly, as a result of corrections, Figures 2G, 2H, and 4B have gained statistical significance (as illustrated in Figures below), strengthening our conclusions and underscoring the unintentional nature of these errors. The corrected Figures 2G, 2H, and 4B, along with the relevant data source, have been appropriately updated.

Figure 2G

Figure 2H

Figure 4B

- Please resubmit source data as one zip file per figure.

Done

- Our production/data editors have asked you to clarify several points in the figure legends:

o Please note that a separate 'Data Information' section is required in the legends of figures 1h; 2b, d-i; 3b, d-g; 4a-d; 5a-b, d-e; 6b-c, e; 7a-d, f-h; 8b, d, f-g; EV 1a-b, g; EV 2c, e; EV 3c-d; EV 4e-g; EV 5c-e. Please see <https://www.embopress.org/page/journal/14693178/authorguide#figureformat> for an example.

Done.

o **Please note that the legend for figure 3b is incorrectly labelled as 3a in the legend. This needs to be rectified.**

The legend labelling is now correct.

o **Please note that the legend for figures 3d-f is incorrectly labelled as 3b-d. This needs to be rectified.**

The legend labelling is now correct.

o **Please indicate the statistical test used for data analysis in the legends of figure 6a; EV 4a, c.**

Done.

o **Please note that in figure 3b; there is a mismatch between the annotated p values in the figure legend and the annotated p values in the figure file that should be corrected.**

Done.

o **Please note that information related to n is missing in the legends of figures 3b; EV 4e-g.**

Done, but please note that EV 4E-G are now called EV4 F-H

o **Although 'n' is provided, please describe the nature of entity for 'n' in the legend of figure 8d.**

Done

o **Please note that the error bars are not defined in the legends of figures 3b; 7h; 8d.**

Done.

o **Please note that the scale bar needs to be defined for figure 1b.**

Done

• ***The manuscript sections should be in the following order: Title page - Abstract & Keywords - Introduction - Results - Discussion - Methods - Data Availability - Acknowledgments - Disclosure Statement & Competing Interests - References - Figure Legends - Tables with legends - Expanded View Figure Legends.***

Done.

• ***The synopsis image needs to be 550px wide and 300-600px high. When your synopsis image is resized accordingly, the labels are too small to read (please see attached). Please provide a synopsis image with larger labels.***

Done.

Prof. Sussan Nourshargh
Queen Mary University of London
Centre for Inflammation and Therapeutic Innovation
London
United Kingdom

Dear Sussan,

Thank you for submitting your revised manuscript. I have now looked at everything and all is fine. Therefore, I am very pleased to accept your manuscript for publication in EMBO Reports.

Congratulations on a nice work!

Kind regards,

Deniz
--
Deniz Senyilmaz Tiebe, PhD
Editor
EMBO Reports
